# Proximity-dependent mapping of the HCMV US28 interactome identifies RhoGEF signaling as a requirement for efficient viral reactivation

**Samuel Medica[1], Lindsey B. Crawford[1¤], Michael Denton[1], Chan-Ki Min[2], Taylor A. Jones[1], Timothy Alexander[1], Christopher J. Parkins[1], Nicole L. Diggins[1], Gabriel J. Streblow[1], Adam T. Mayo[1], Craig N. Kreklywich[1], Patricia Smith[1], Sophia Jeng[3], Shannon McWeeney[3], Meaghan H. Hancock[1], Andrew Yurochko[2], Michael S. Cohen[4], Patrizia Caposio[1], Daniel N. Streblow[1,5]***

**1** Vaccine and Gene Therapy Institute, Oregon Health and Science University, Beaverton, Oregon, United States of America, **2** Department of Microbiology & Immunology, Center for Molecular & Tumor Virology, Louisiana State University Health Sciences Center-Shreveport, Shreveport, Louisiana, United States of America, **3** Department of Bioinformatics and Computational Biology, Oregon Health and Science University, Portland, Oregon, United States of America, **4** Department of Chemical Physiology and Biochemistry, Oregon Health and Science University, Portland, Oregon, United States of America, **5** Division of Pathobiology and Immunology, Oregon National Primate Research Center, Beaverton, Oregon, United States of America

¤ Current address: Department of Biochemistry, University of Nebraska–Lincoln, Lincoln, Nebraska, United States of America

* streblow@ohsu.edu

**Data Availability Statement:** All relevant data are within the manuscript and its Supporting Information files.

## Abstract

Human cytomegalovirus (HCMV) encodes multiple putative G protein-coupled receptors (GPCRs). US28 functions as a viral chemokine receptor and is expressed during both latent and lytic phases of virus infection. US28 actively promotes cellular migration, transformation, and plays a major role in mediating viral latency and reactivation; however, knowledge about the interaction partners involved in these processes is still incomplete. Herein, we utilized a proximity-dependent biotinylating enzyme (TurboID) to characterize the US28 interactome when expressed in isolation, and during both latent (CD34$^+$ hematopoietic progenitor cells) and lytic (fibroblasts) HCMV infection. Our analyses indicate that the US28 signalosome converges with RhoA and EGFR signal transduction pathways, sharing multiple mediators that are major actors in processes such as cellular proliferation and differentiation. Integral members of the US28 signaling complex were validated in functional assays by immunoblot and small-molecule inhibitors. Importantly, we identified RhoGEFs as key US28 signaling intermediaries. *In vitro* latency and reactivation assays utilizing primary CD34$^+$ hematopoietic progenitor cells (HPCs) treated with the small-molecule inhibitors Rhosin or Y16 indicated that US28 –RhoGEF interactions are required for efficient viral reactivation. These findings were recapitulated *in vivo* using a humanized mouse model where inhibition of RhoGEFs resulted in a failure of the virus to reactivate. Together, our data identifies multiple new proteins in the US28 interactome that play major roles in viral latency and reactivation, highlights the utility of proximity-sensor labeling to characterize protein interactomes, and provides insight into targets for the development of novel anti-HCMV therapeutics.

**Funding:** This work was supported by a grant from the National Institutes of Health NIAID (P01 AI127335, DNS). The funders had no role in the study design, data collection, analysis of results, decision to publish, or preparation of the manuscript.

**Competing interests:** The authors have declared that no competing interests exist.

## Author summary

Human cytomegalovirus (HCMV), continues to be amongst the most prevalent viral infections worldwide. Primary infection of HCMV is often asymptomatic and results in the establishment of latency within cells of myeloid lineage. Once latency is established, the virus will persist throughout the host's lifetime. Subsequent viral reactivation events can pose life-threatening health complications for the immunocompromised population; including transplant recipients and AIDS patients. Many factors have been shown to mediate the switch from latent to lytic HCMV infection such as signal transduction through the viral G protein-coupled receptor (vGPCR) US28. In the present report, we utilize proximity-dependent labeling coupled with mass spectrometry to identify host and viral proteins proximal to US28. Our analysis indicates significant overlap between US28 and the EGFR and RhoA signaling pathways. We further explored the relationship between US28 and the RhoA signal transduction pathway to identify RhoGEFs as an important member of the US28 signalosome. Our data indicates that ablation of RhoGEF activity significantly attenuates US28 signaling. Furthermore, we show that pharmacological inhibition of RhoGEFs results in an inability of the virus to efficiently reactivate *in vitro* and *in vivo*. These findings reveal previously unknown US28 interactors, which play an integral role in the facilitation of viral reactivation, and provide the first example of specific cellular factors being implicated in US28 function and viral reactivation *in vivo*.

## Introduction

Human cytomegalovirus (HCMV) is the largest member of the β-herpesvirus family and infects the majority of the world population [1,2]. The virus persists as a lifelong infection through latency establishment in hematopoietic progenitor cells (HPCs) located in the bone marrow [3]. Latently infected monocytes generated from these HPCs are thought to be the cellular reservoir [4–6] from which the virus disseminates to other tissues of the body. Viral reactivation events pose a major risk during solid organ and bone marrow transplantation and can lead to CMV-associated disease including organ failure and graft rejection [7–14]. Several cellular signaling pathways have been implicated to be involved with HCMV latency and reactivation, including EGFR, PI3K/AKT, MAPK, TGF-β, Src, ERK, Rho, and Wnt pathways [15–22]; however, the exact signaling mechanisms that contribute to the establishment of latency and potential to reactivate remain unclear. Moreover, current FDA-approved HCMV antivirals often have toxic effects and primarily target late phases of viral replication when clinical manifestations are already present. Therefore, in order to discover additional treatment options for HCMV, it is crucial that we elucidate the molecular mechanisms mediating viral latency and reactivation.

HCMV encodes four putative G protein-coupled receptors (GPCRs) with homology to cellular chemokine receptors; however, US28 has been the most extensively characterized to date. US28 is expressed in infected human peripheral blood cells during periods of latency [23] and during reactivation episodes [24, 25]. US28 signaling results in the activation of multiple transcription factors involved in cellular proliferation, differentiation, and migration; including nuclear factor of activated T cells (NF-AT), cAMP-response element binding protein (CREB), nuclear factor kappa-light chain enhancer of activated B cells (NF-κB), serum response factor (SRF), signal transducer and activator of transcription 3 (STAT3), and β-catenin [15–22]. We, and others, have previously demonstrated that US28 is required for HCMV reactivation in latently infected CD34+ HPCs [26–28] and that US28 drives cellular differentiation down the

myeloid lineage in HCMV-infected CD34[+] HPCs [28]. Additionally, our previous studies show that US28 is required for both the maintenance of viral latency and the capacity to reactivate *in vivo* utilizing a humanized mouse model [28]. Combined, these data indicate that US28 is required for latency/reactivation *in vitro* and *in vivo*; however, the specific cellular signaling pathways involved have yet to be defined.

US28 uniquely binds both CC chemokines (RANTES, MCP-1, MIP-1α) and $CX_3C$-chemokines (Fractalkine) [29–31]. Depending on the ligand stimulus and infected cell type, US28 signaling can result in the activation of multiple signal transduction pathways. For instance, US28 coupling with $G\alpha_{12/13}$ proteins, and subsequent activation of RhoA and downstream effector Rho-associated kinase (ROCK) is critical for promoting actin reorganization and cellular migration in infected smooth muscle cells and monocytes [32,33]. Cell migration, differentiation, and other cellular processes are tightly regulated in part by activation of Rho GTPases, which are in turn regulated by Rho guanine nucleotide exchange factors (RhoGEFs) [34,35]. GEFs provide a direct link between the activation of RhoA and the cell-surface receptors for growth factors (i.e., EGFR), cytokines and chemokines (i.e., RANTES, MCP-1, MIP-1α), and G protein-coupled receptors (i.e., US28). Because cellular differentiation and migration are essential for the switch from latent to lytic HCMV infection, RhoGEFs may serve as key regulators of cellular signaling pathways involved in viral latency and reactivation. However, it is difficult to determine the consequences of US28 signaling without a complete understanding of the protein interactions that occur during signal transduction.

In the current study, we utilized an unbiased proximity-dependent labeling enzyme (TurboID) to characterize proteins that are proximal to US28 in multiple relevant *in vitro* cell models. Our proteomic analysis identified multiple novel proteins involved in US28 signal transduction. We further explored the relationship between US28 and the RhoA signal transduction pathway to identify RhoGEFs as important facilitators of viral reactivation. Our data indicates that ablation of RhoGEF activity, via pharmacological inhibition, attenuates US28 signaling activity. Furthermore, we show that inhibition of RhoGEFs, via the small-molecule inhibitors Rhosin and Y16, impedes efficient viral reactivation in primary CD34[+] HPCs. Utilizing a humanized NSG mouse model, we show that treatment with Rhosin resulted in failure of the virus to reactivate. Collectively, our data demonstrates that RhoGEFs are integral components of the US28 signalosome and are required for efficient viral reactivation.

## Results

### Characterization of US28-TurboID constructs

Our understanding of US28 signaling pathways and the proteins that act in concert with US28 is still incomplete. To characterize how US28 signaling influences latency, reactivation, and lytic infection, we took an unbiased approach to determine interactors within the US28 signalosome. By affixing TurboID to the C' terminal tail of US28, we developed a system to assess the US28 interactome directly in living cells. Expression of the HA-tagged US28-TurboID protein (HA-US28-BT), at the expected size, was verified by immunoblot after transfection into HEK293M cells (**Fig 1A**). We chose to use HEK293M cells because of their efficient transfectability and prior use in US28 functional assays. Next, we examined the capability of HA-US28 and HA-US28-BT to transcriptionally activate the reporter elements SRE and SRF in transiently transfected HEK293M cells. In the absence of any exogenous ligands, transfected HA-US28 and HA-US28-BT were able to stimulate $G\alpha_{q/11}$ and downstream SRE at levels several-fold above transfection with the empty pcDNA3.1 vector alone (**Fig 1B**). In a similar manner, transfected HA-US28 and HA-US28-BT induced activation of $G\alpha_{12/13}$ and downstream RhoA signaling as measured via SRF reporter element activation (**Fig 1C**). To confirm that

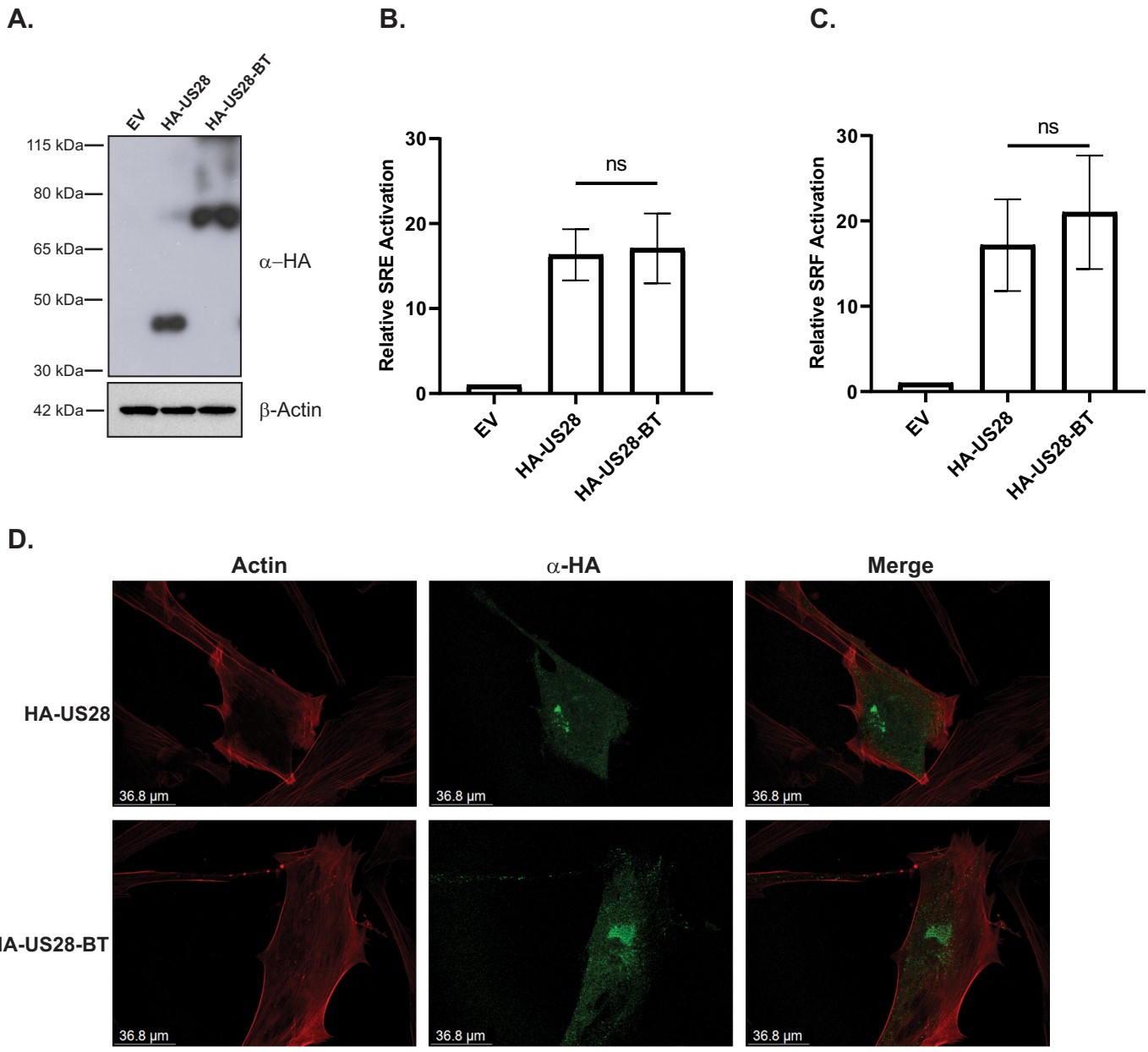

**Fig 1. Expression, Signaling, and Localization of US28 BioID Constructs. (A)** HEK293M cells were transfected with pcDNA3.1-HA-US28 (HA-US28), pcDNA3.1-HA-US28-TurboID (HA-US28-BT), or the empty pcDNA3.1 vector (EV). Lysates were harvested 24 hours post-transfection and expression was confirmed via immunoblot using an anti-HA antibody (n = 3, representative blot shown). **(B & C)** HEK293M cells were transfected with HA-US28, HA-US28-BT, or the empty pcDNA3.1 vector (EV) along with Renilla and luciferase reporter plasmids for **(B)** SRE or **(C)** SRF. At 18 hours post-transfection, media was exchanged with serum-free DMEM. Luciferase activity was measured using the Dual-Luciferase Reporter Assay System (Promega) at 6 hours post-media exchange. Error bars represent the standard error of the mean between triplicate experiments. Statistical significance was calculated by one-way ANOVA followed by Tukey's multiple comparisons test between experimental groups. **(D)** NHDF cells were seeded onto coverslips and transfected with either HA-US28 or HA-US28-BT. At 48 hours post-transfection, cells were fixed, permeabilized, and stained overnight against HA (green) and phalloidin (actin-red) (n = 2, representative images shown).

addition of TurboID does not alter localization of US28, we performed immunofluorescence imaging analysis on human fibroblasts transfected with HA-US28 and HA-US28-BT. No discernable difference in localization was observed between the two constructs **(Fig 1D)**.

Together, these results show that HA-US28-BT is efficiently expressed in HEK293M cells and that the addition of TurboID to US28 does not impact signaling activity or localization.

## Identification of the US28 interactome

Next, we sought to confirm that our system could efficiently label proteins within the US28 interactome, including those with transient or short-lived interactions (**Fig 2A**). We chose to use HEK293M cells for initial characterization of the interactome before validating results in infected cells. To this end, HEK293M cells were transfected with HA-US28 and HA-US28-BT. At 18 hours post transfection, the culture medium was supplemented with biotin and lysates harvested six hours thereafter. The resulting tagged proteins were purified via streptavidin mediated bead-based precipitation. One quarter of the same purified protein lysate used for mass spectrometry was analyzed to verify input and control conditions. Coomassie staining verified comparable protein content in HA-US28 and HA-US28-BT transfected whole cell lysates (**Fig 2B, right panel: Whole Lysate**). Moreover, appreciable amounts of protein were only detectable in cells transfected with HA-US28-BT after streptavidin-mediated purification and pulldown of biotinylated proteins (**Fig 2B, left panel: SA-PD**). Further analysis by immunoblot using HRP-conjugated streptavidin confirms efficient labeling (**Fig 2B, left panel: SA-PD**) and specific pulldown (**Fig 2B, right panel: Unbound**). While transient transfection in established cell lines is a tractable model for initial studies, the differential cellular signaling events that occur during the course of infection are not accurately captured in these systems. To characterize the host and viral proteins that interact with US28 during infection, we engineered a recombinant virus using the TB40/E-GFP backbone and affixing the TurboID enzyme onto the C' terminal tail of US28 (TB40/E-GFP-US28-BT). NHDF or human embryonic stem cell (hESC) -derived CD34+ HPCs were mock infected, or infected with TB40/E-GFP or TB40/E-GFP-US28-BT at a MOI of 2. In NHDF experiments, the culture medium was supplemented with biotin at 3-days post infection (dpi) and cell lysates were harvested six hours thereafter. CD34+ HPCs were cultured in conditions that promote latent infection as previously described [28]. At 14-dpi, the culture medium was supplemented with biotin and cells were incubated overnight prior to cell lysis. Viral infection and efficient biotin ligation were confirmed in whole cell lysates derived from NHDFs via immunoblot using HRP-conjugated streptavidin and antibodies directed against HCMV pUL44 and TurboID (**Fig 2C**). Similar results were obtained using whole cell lysates derived from CD34+ HPCs (**Fig 2D**). The resulting interacting proteins for all three *in vitro* models were purified via streptavidin bead-based precipitation, and analyzed by liquid chromatography tandem mass spectrometry (LC-MS/MS).

## US28 signals through multiple pathways including RhoGEFs

Hits from our proteomic analysis were refined by excluding proteins that were identified as likely contaminants based on comparison to the CRAPome database [36]. Our revised list included 984, 1,054, and 843 host proteins which were in close proximity to US28 in HEK293M, NHDF, and CD34+ HPC datasets, respectively. To examine the cellular signaling pathways whose components were in close proximity to US28, we used the pathway analysis tool Reactome [37] to identify significantly enriched signal transduction pathways (**Fig 3A–3C and S1–S3 Tables**). Our analysis indicated enrichment of proteins that are members of multiple cellular signaling pathways contributing to membrane trafficking, cellular metabolism, differentiation, and cellular migration. Consistent with other studies, our pathway analysis of the US28 signaling complex in all three cell culture models demonstrates significant overlap between US28 and the RhoA signaling pathway [32,33,38,39]. While we identified EphA2 in

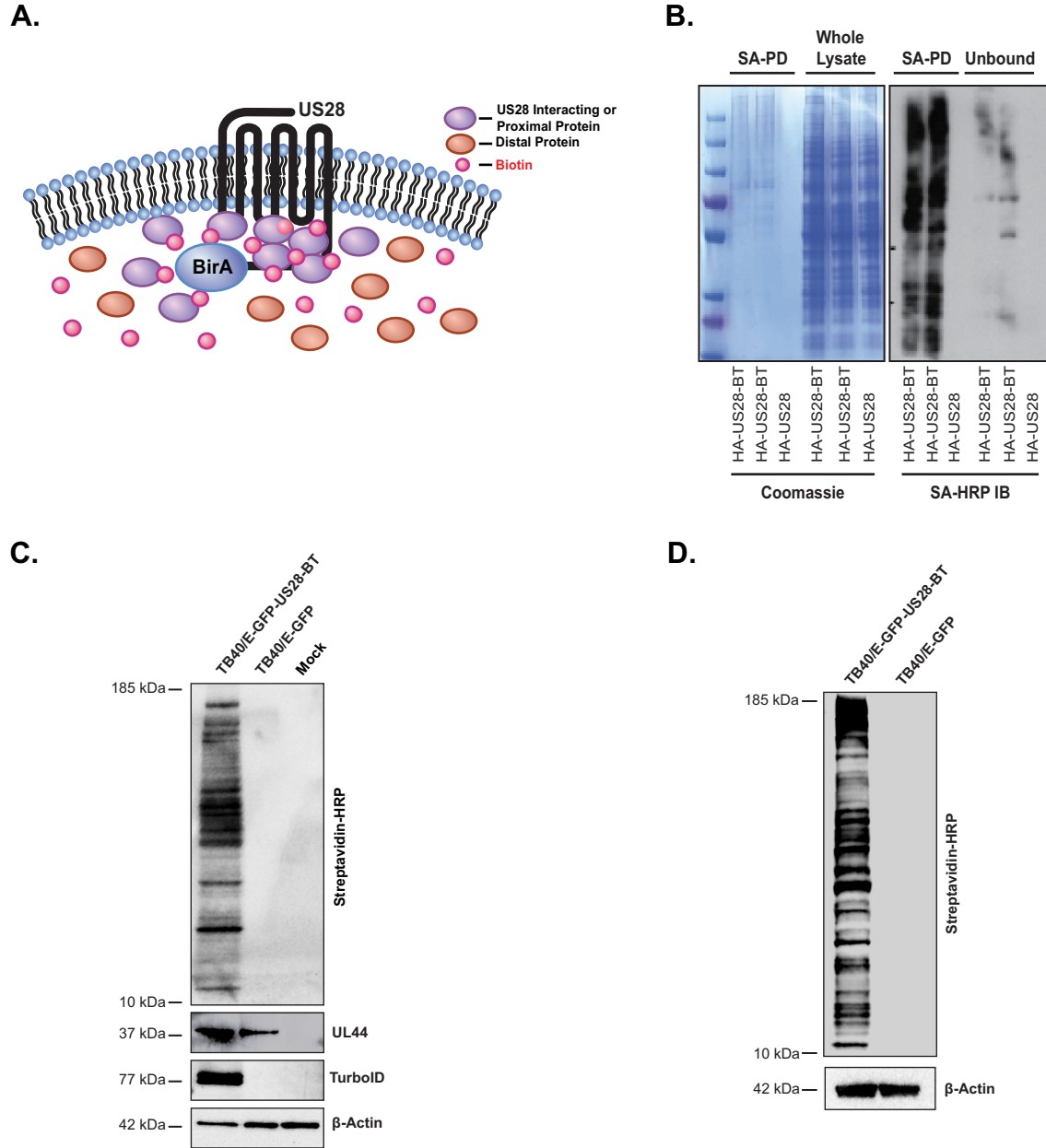

**Fig 2. Labeling and Purification of Transfected and Infected Lysates. (A)** Experimental outline of US28-TurboID labeling in cells. Interacting or proximal proteins (purple) are tagged with biotin (red) while distal proteins (orange) remain untagged. Tagged proteins are purified and analyzed by mass spectrometry to identify proteins that are in close proximity to US28. **(B)** HEK293M cells were transfected with pcDNA3.1-HA-US28-BT (HA-US28-BT) or pcDNA3.1-HA-US28 (HA-US28). At 18 hours post-transfection the cell culture medium was supplemented with biotin (50μg/mL) for 6 hrs. Tagged proteins were bound to NeutrAvidin beads and incubated overnight prior to extensive washing, trypsin digestion, and formic acid treatment. Protein content, purification, and efficient labeling were confirmed by Coomassie staining and streptavidin-specific immunoblot (n = 2, representative images shown). **(C)** NHDF cells or **(D)** CD34[+] HPCs were infected with TB40/E-GFP-US28-BT or TB40/E-GFP at a MOI of 2. At 3 dpi, the cell culture medium was supplemented with biotin (50μg/mL) for 6 hrs. Labeling and infection were confirmed by immunoblot on whole cell lysates using the indicated primary antibodies. Purified proteins were analyzed by mass spectrometry for protein identification (n = 3, representative blot shown).

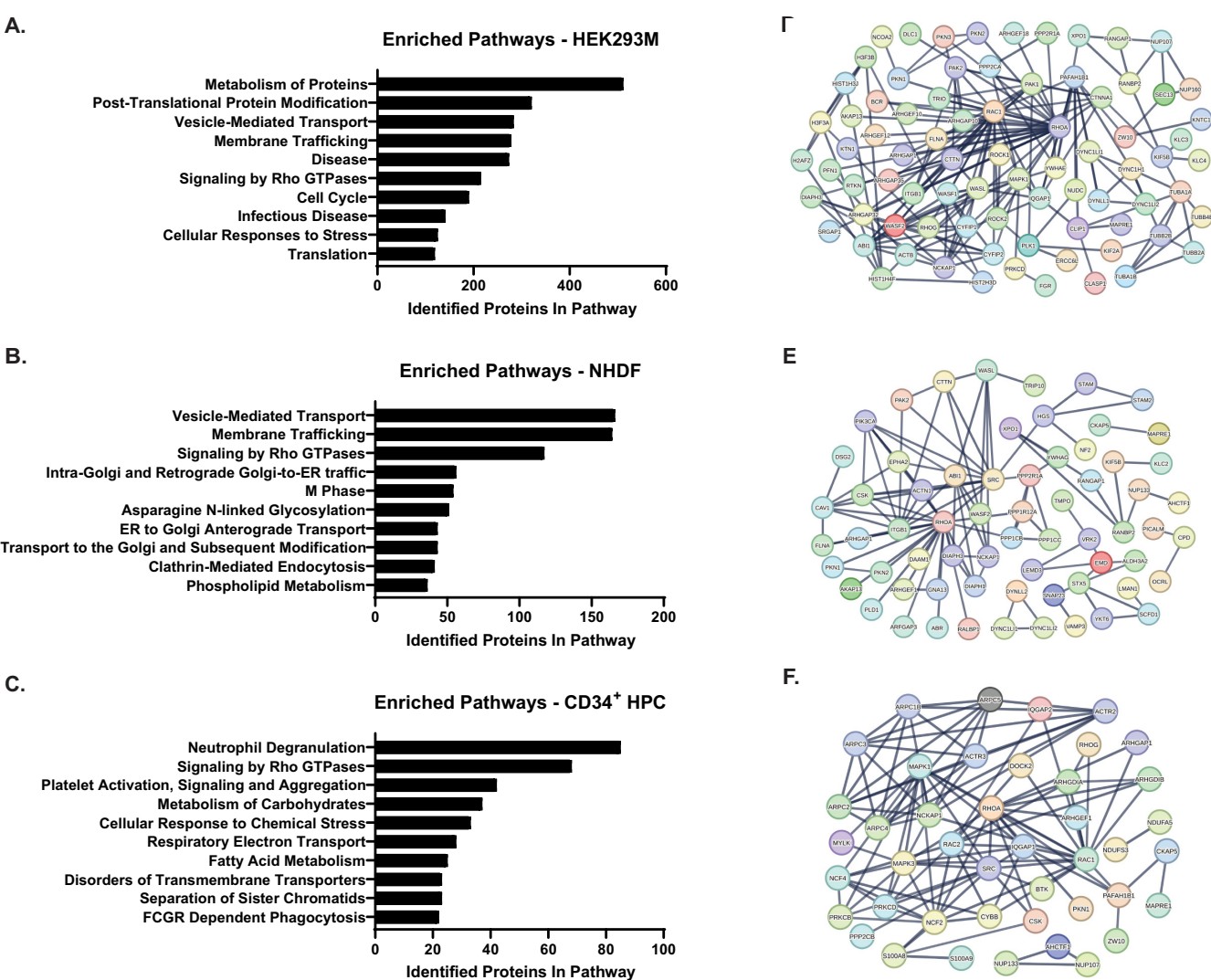

**Fig 3. Network Analysis of the US28 Interactome.** Proteins within the US28 interactome were identified via LC-MS/MS and analyzed using Reactome and STRING software tools. **(A-C)** Proteins identified by LC-MS/MS were analyzed using Reactome for **(A)** HEK293M cells transfected with pcDNA3.1-HA-US28-BT, **(B)** NHDFs infected with TB40/E-GFP-US28-BT, and **(C)** CD34+ HPCs infected with TB40/E-GFP-US28-BT. Significantly enriched pathways and the number of associated proteins were plotted using GraphPad Prism 9.0 software. **(D-F)** STRING network analysis of the US28 interactome identified RhoGEFs as members of the US28 interactome in **(D)** HEK293M cells transfected with pcDNA3.1-HA-US28-BT, **(E)** NHDFs infected with TB40/E-GFP-US28-BT, and **(F)** CD34+ HPCs infected with TB40/E-GFP-US28-BT.

the US28 interactome during proximity-dependent labeling in NHDFs, which was consistent with previous reports [39,40], we did not detect this cellular protein tyrosine signaling molecule in the US28 interactome in CD34+ HPCs but we did consistently detect Src (**Fig 4A**) [41]. Because we have previously shown that US28 signaling stimulates $G\alpha_{12/13}$ activity [32], we decided to further explore US28 –RhoA interactions using the STRING database to map identified interactors and their downstream effectors. Interestingly, Rho-specific guanine nucleotide exchange factors (GEFs) responsible for activation of RhoA, were highly enriched in the US28 proximity labeling in all three cell types/conditions (**Fig 3D–3F**). In addition to host proteins, we also identified several viral proteins that are potential interaction partners of US28 during both latent and lytic infection. In cellular lysates obtained from infected NHDFs, we identified 28 viral proteins that may interact with US28 (**Fig 4B and Table 1**). Interestingly,

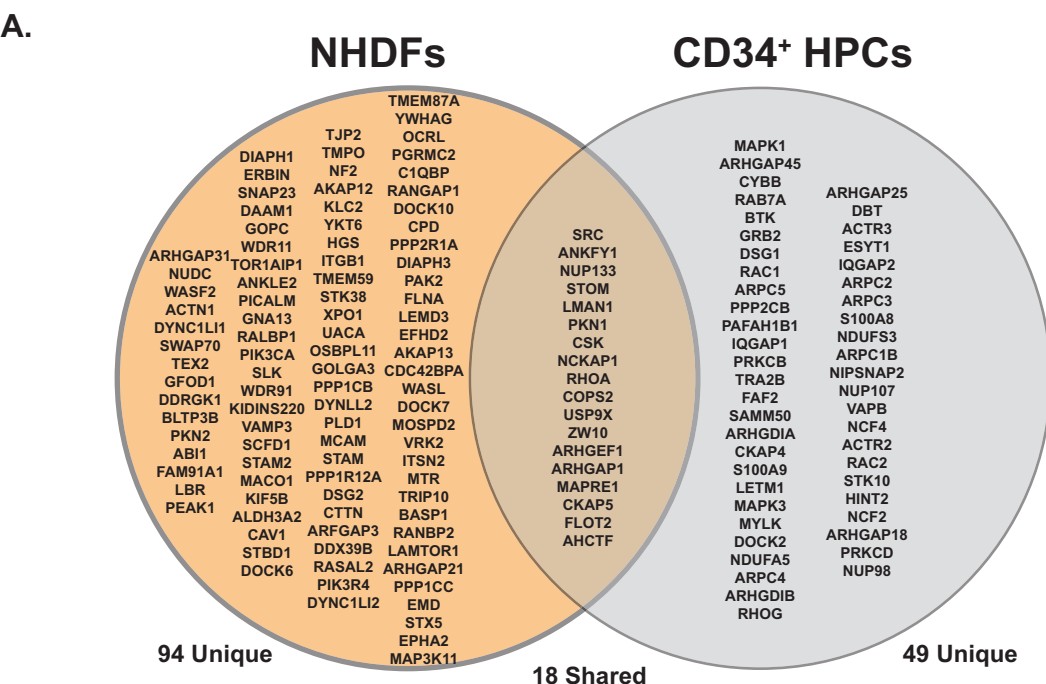

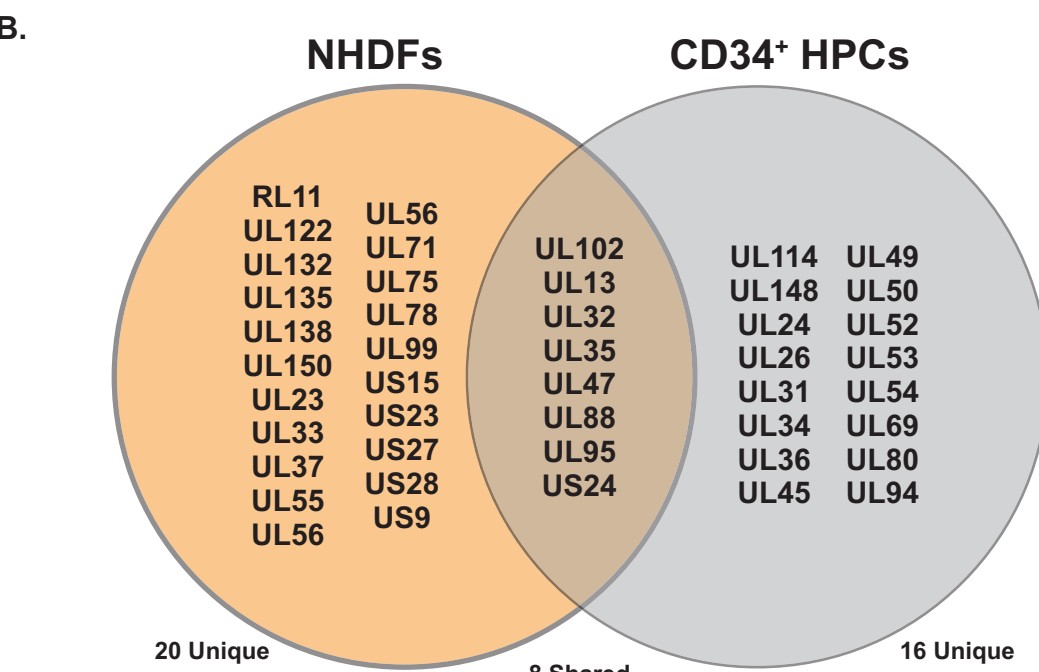

**Fig 4. Comparison of the US28 –Rho GTPase Specific Interactomes During Latent and Lytic Infection. (A)** Candidate host US28 interaction partners specific to Rho GTPase signaling were compared between NHDFs and latently infected CD34+ HPCs. **(B)** Candidate viral US28 interaction partners were compared between NHDFs and latently infected CD34+ HPCs.

several viral GPCRs (UL33, UL78, and US27), glycoproteins (UL55, UL75, UL132, and US9), and tegument proteins (UL23, UL35, UL47, UL71, UL88, US23, and US24) were shown to be in close proximity to US28. These data would suggest that, during late lytic replication

**Table 1. Viral Proteins Within the US28 Interactome in Infected NHDF cells.**

| Protein | Description | q-Value | PEP Score | # Peptides | PSMs | AAs | MW [kDa] |
|---|---|---|---|---|---|---|---|
| HCMV RL11 | Unknown | 0.000 | 10.46 | 2 | 4 | 234 | 26.6 |
| HCMV UL102 | Primase-Associated Factor | 0.000 | 46.53 | 10 | 50 | 874 | 94 |
| HCMV UL122 | Major Immediate-Early Transactivator | 0.000 | 39.93 | 8 | 47 | 564 | 61 |
| HCMV UL13 | Mitochondria MICOS stabilizer | 0.000 | 93.84 | 17 | 84 | 473 | 54.5 |
| HCMV UL132 | Glycoprotein; Formation of Assembly Compartment | 0.000 | 183.35 | 19 | 278 | 270 | 29.7 |
| HCMV UL135 | Viral reactivation | 0.000 | 60.26 | 11 | 55 | 308 | 33.3 |
| HCMV UL138 | Viral latency | 0.000 | 18.21 | 4 | 16 | 169 | 19.3 |
| HCMV UL150 | Viral Entry | 0.000 | 39.99 | 6 | 35 | 328 | 35.1 |
| HCMV UL23 | Tegument Protein; Particle Infectivity | 0.000 | 9.35 | 3 | 5 | 284 | 32.9 |
| HCMV UL32 | Tegument Phosphoprotein | 0.000 | 137.07 | 26 | 112 | 1049 | 112.7 |
| HCMV UL33 | Viral G Protein-Coupled Receptor | 0.000 | 48.70 | 6 | 45 | 411 | 46 |
| HCMV UL35 | Tegument Protein; Particle Infectivity | 0.000 | 18.93 | 4 | 16 | 641 | 72.6 |
| HCMV UL37 | Viral Mitochondrion-Localized Inhibitor of Apoptosis | 0.000 | 5.87 | 2 | 12 | 202 | 23.2 |
| HCMV UL47 | Tegument Protein; Particle Infectivity | 0.000 | 46.53 | 11 | 26 | 983 | 110 |
| HCMV UL55 | Glycoprotein B (gB); Viral Entry | 0.000 | 77.01 | 11 | 62 | 907 | 101.9 |
| HCMV UL56 | Viral DNA packaging | 0.000 | 69.53 | 13 | 44 | 850 | 95.7 |
| HCMV UL71 | Tegument Protein; Particle Infectivity | 0.000 | 191.64 | 28 | 252 | 361 | 39.8 |
| HCMV UL75 | Glycoprotein H (gH); Viral Infectivity | 0.000 | 26.19 | 6 | 19 | 743 | 84.4 |
| HCMV UL78 | Viral G Protein-Coupled Receptor | 0.000 | 26.47 | 5 | 21 | 431 | 47.3 |
| HCMV UL88 | Tegument Protein | 0.000 | 45.67 | 12 | 38 | 429 | 47.7 |
| HCMV UL95 | Late Gene Expression | 0.000 | 27.94 | 4 | 16 | 531 | 57.2 |
| HCMV UL99 | Tegument Phosphoprotein | 0.000 | 6.88 | 2 | 9 | 190 | 20.9 |
| HCMV US15 | Tegument Protein | 0.000 | 4.72 | 2 | 5 | 262 | 29.1 |
| HCMV US23 | Tegument Protein, Viral Transactivator | 0.000 | 8.35 | 2 | 9 | 592 | 68.9 |
| HCMV US24 | Unknown | 0.000 | 27.78 | 7 | 15 | 501 | 58 |
| HCMV US27 | Viral G Protein-Coupled Receptor | 0.001 | 3.94 | 2 | 9 | 361 | 41.9 |
| HCMV US28 | Viral G Protein-Coupled Receptor | 0.000 | 42.58 | 8 | 43 | 354 | 41 |
| HCMV US9 | Glycoprotein, blocks IFN signaling | 0.000 | 7.02 | 3 | 7 | 247 | 28 |

conditions, US28 is associated with the viral assembly compartment. In lysates obtained from latently infected CD34+ HPCs, we identified 24 viral proteins that interact with, or are within close proximity, to US28 (**Fig 4B** and **Table 2**). In contrast to our analysis in infected NHDFs, proteins within the latency (CD34+ HPC) dataset were involved in immune evasion (UL31) and gene expression (UL34, UL102, UL54, UL69, and UL95). A comparison between US28 interactomes in infected NHDFs and CD34+ HPCs yielded 18 shared host and 8 shared viral proteins (**Fig 4**). Previous pulldown analyses of the US28 interactome identified 47 HCMV proteins in lysates from human fibroblasts and of these we detected 28 in NHDFs during lytic replication [39]. An additional 10 viral proteins were identified in our proximity labeling in CD34+ HPCs that were unique to this cell type but overlapped with those reported previously [39]. Interestingly, UL69 was the most highly represented viral protein in our CD34+ HPC US28 interactome, we also detected the UL69 interacting protein Suppressor of Ty 6 and other elongation factors associated with UL69 [42] but not the mRNA export protein U2AF65 [43] indicating that US28 may form a complex linked with UL69. Thus, our US28 proximity labeling results largely overlap with the viral and cellular proteins reported previously, and we demonstrate a consistent intersection of US28 and the Rho signaling pathways during both latent and lytic phases of the virus lifecycle.

**Table 2. Viral Proteins Within the US28 Interactome in Latently Infected CD34⁺ HPCs.**

| Protein | Description | q-Value | PEP Score | # Peptides | PSMs | AAs | MW [kDa] |
|---------|-------------|---------|-----------|------------|------|-----|----------|
| HCMV UL102 | Primase-Associated Factor | 0.000 | 15.977 | 6 | 11 | 874 | 94 |
| HCMV UL114 | DNA Repair | 0.000 | 18.776 | 4 | 6 | 250 | 28.3 |
| HCMV UL13 | Mitochondria MICOS Stabilizer | 0.000 | 3.643 | 2 | 2 | 473 | 54.5 |
| HCMV UL148 | Chaperone Protein | 0.000 | 11.065 | 5 | 14 | 316 | 36.4 |
| HCMV UL24 | Unknown | 0.000 | 10.054 | 3 | 5 | 300 | 34.2 |
| HCMV UL26 | Tegument Protein | 0.000 | 10.021 | 4 | 8 | 222 | 24.9 |
| HCMV UL31 | Innate Immune Evasion | 0.000 | 32.035 | 11 | 17 | 595 | 65.6 |
| HCMV UL32 | Tegument Phosphoprotein | 0.000 | 24.99 | 8 | 12 | 1049 | 112.7 |
| HCMV UL34 | Transcriptional Repressor | 0.000 | 32.069 | 7 | 19 | 407 | 45.4 |
| HCMV UL35 | Tegument Protein; Particle Infectivity | 0.000 | 43.213 | 12 | 20 | 641 | 72.6 |
| HCMV UL36 | Tegument Protein | 0.000 | 19.365 | 6 | 13 | 453 | 52.2 |
| HCMV UL45 | Unknown | 0.000 | 30.066 | 12 | 16 | 906 | 101.7 |
| HCMV UL47 | Tegument Protein; Particle Infectivity | 0.000 | 15.556 | 4 | 6 | 983 | 110 |
| HCMV UL49 | Viral Pre-Initiation Complex | 0.000 | 4.307 | 2 | 3 | 570 | 63.7 |
| HCMV UL50 | Viral Egress | 0.000 | 39.422 | 12 | 22 | 400 | 43.2 |
| HCMV UL52 | Capsid Localization | 0.000 | 11.506 | 4 | 8 | 667 | 74.1 |
| HCMV UL53 | Viral Egress | 0.000 | 3.265 | 2 | 2 | 376 | 42.3 |
| HCMV UL54 | Viral Polymerase | 0.000 | 7.739 | 3 | 4 | 1242 | 137.1 |
| HCMV UL69 | Transcriptional Activator | 0.000 | 21.081 | 6 | 13 | 741 | 82.3 |
| HCMV UL80 | Scaffold Protein | 0.000 | 13.709 | 5 | 10 | 708 | 73.7 |
| HCMV UL88 | Tegument Protein | 0.000 | 8.695 | 3 | 6 | 429 | 47.7 |
| HCMV UL94 | Viral Egress | 0.000 | 17.437 | 5 | 16 | 345 | 38.2 |
| HCMV UL95 | Late Gene Expression | 0.000 | 8.086 | 3 | 5 | 531 | 57.2 |
| HCMV US24 | Tegument Protein | 0.000 | 12.056 | 4 | 4 | 501 | 58 |

## Inhibition of RhoGEFs attenuates US28 signaling

To validate our analyses, we took a multifaceted approach to confirm select cellular interaction partners of US28, and to examine the effects of these interactions on US28 signaling. First, we confirmed the association of previously identified host proteins that interact with US28. Our group, and others, have shown that US28 interacts with Src and ERK mediating several signal transduction events [33,38,41,44]. The presence of newly identified interactors PDZ-RhoGEF, p115-RhoGEF, and ROCK1, as well as the known interactors (Src and ERK), were validated by traditional immunoblot following streptavidin pulldown on lysates harvested from NHDF cells infected with TB40/E-GFP-US28-BT (**Fig 5A**). Next, to examine the effects of RhoGEF interactions on US28 signaling, we identified two small-molecule compounds inhibiting various aspects of RhoGEF signaling. The small molecule inhibitors, Rhosin and Y16, sterically block RhoA interactions with associated GEFs [45,46], and inhibited RhoA activation in a dose-dependent manner (**S1A Fig**). In US28 signaling assays, treatment with either Rhosin or Y16 lead to a 70–72% and 53–60% reduction in US28-mediated activation of the SRF reporter element, respectively (**Fig 5B**). Taken together, these results indicate a robust association between US28 and the RhoA signaling pathway during both lytic and latent phases of the viral lifecycle.

## RhoGEFs are required for efficient reactivation of HCMV from latency

Because we identified and validated RhoGEFs in all of our US28 interactome models, we examined the extent of RhoGEF involvement in viral latency and reactivation. To this end, we

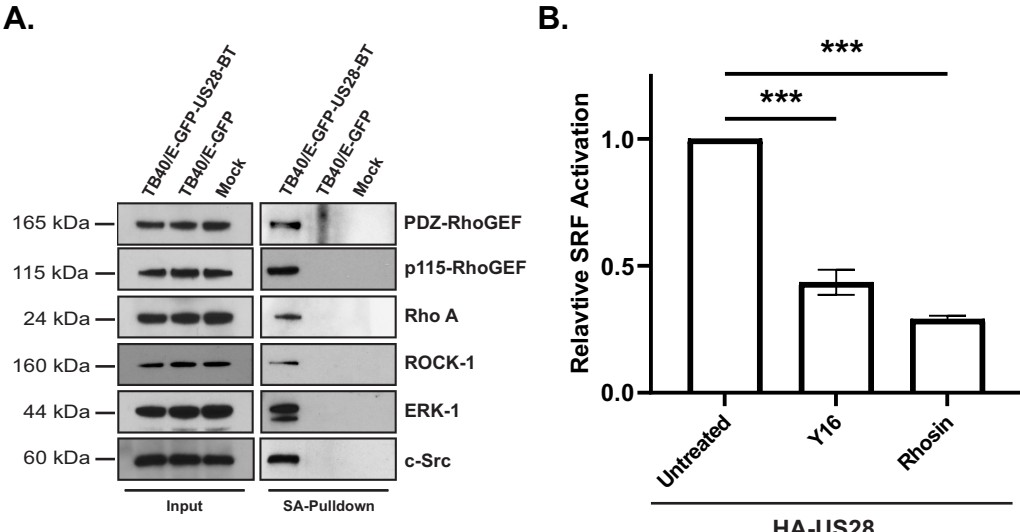

**Fig 5. Validation of US28 Interactome Analysis. (A)** NHDF cells were mock infected, or infected with TB40/E-GFP-US28-BT or TB40/E-GFP at a MOI of 2. At 3 DPI, the cell culture medium was supplemented with biotin (50μg/mL) for 6 hrs. Lysates were harvested and tagged proteins were bound to NeutrAvidin beads and incubated overnight prior to extensive washing. The presence of indicated proteins was confirmed via traditional immunoblot using the indicated primary antibodies (n = 3, representative blots shown). **(B)** HEK293M cells were transfected with pcDNA3.1-HA-US28 (HA-US28) or the empty pcDNA3.1 vector along with Renilla and SRF reporter plasmids. At 18 hours post-transfection, media was changed to serum-free DMEM supplemented with Rhosin (5μM) or Y16 (5μM). Luciferase activity was measured using the Dual-Luciferase Reporter Assay System (Promega) 6 hours post-media replacement. Error bars represent the standard error of the mean between triplicate experiments and statistical significance was calculated by one-way ANOVA followed by Tukey's multiple comparisons test between experimental groups. $P$ values are listed for significant comparisons where *** $P < 0.001$.

further characterized the pharmacological inhibitors Rhosin and Y16. *In vitro* cytotoxicity experiments for both compounds showed limited to no deleterious effects on cell survival at 72 hours post-treatment in NHDFs and CD34$^+$ HPCs (**S1B and S1C Fig**). In addition to cytotoxicity experiments, we confirmed that neither Rhosin nor Y16 influenced viral replication in fibroblasts using the reporter virus TB40/E-GFP-gHnLuc, which expresses nano-luciferase under the rhesus cytomegalovirus gH viral promoter. Results from this experiment showed no significant effect on viral replication at concentrations as high as 40μM when compared to untreated control cells and cells treated with the HCMV antiviral Foscarnet, which demonstrated robust antiviral activity (**S1D Fig**).

To determine if RhoGEF activity is required for reactivation of HCMV in progenitor cells, CD34$^+$ HPCs were isolated from four independent primary donors, infected with TB40/E-GFP, and cultured to establish latency as previously described [28]. Confirmation that RhoGEFs targeted by Rhosin and Y16 were sufficiently expressed in our CD34$^+$ HPC system was accomplished by RT-qPCR in uninfected cells (**S2 Fig**). To block RhoGEF interactions during reactivation, latently infected HPCs were plated on a fibroblast layer in reactivation supportive media containing either Rhosin or Y16. The establishment of latency was confirmed in each donor by virus production under conditions that promote reactivation combined with the absence of infectious virus particles in latently infected cells (pre-reactivation). Treatment with Rhosin, to block RhoA interaction with upstream GEFs, results in a 36–49% decrease in the amount of virus produced during reactivation (**Figs 6 and S3**). Using Y16 to target RhoA–LARG, p115-RhoGEF, and PDZ-RhoGEF interactions, results in a comparable (31–47%) decrease in the amount of virus produced during reactivation (**Figs 6 and S3**). Together, these data show that RhoGEF activity is required for efficient viral reactivation in CD34$^+$ HPCs.

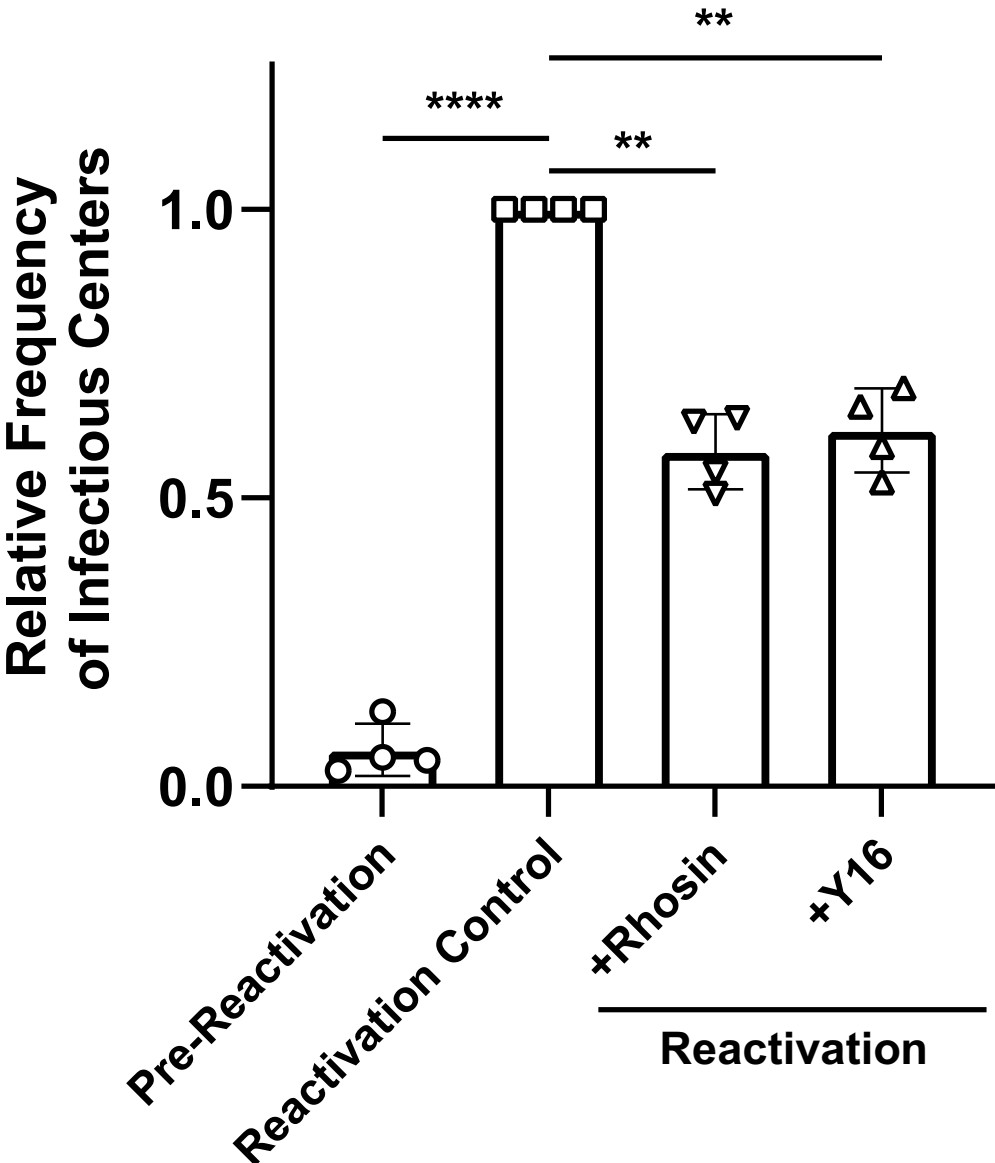

**Fig 6. RhoGEF Interactions Contribute to HCMV Reactivation.** Primary CD34+ hematopoietic progenitor cells (HPCs) isolated from four independent donors were infected with TB40/E-GFP (WT-HCMV), and then FACS isolated for viable, CD34+, GFP+ HPCs at 2 dpi as previously described [28]. Infected HPCs were cultured on stromal cell support for 12 days to establish latency and equivalent populations of HPCs were co-cultured in supportive media on a fibroblast layer to reactivate virus as previously described [28]. Samples were treated with the RhoGEF inhibitors Rhosin (10nM) or Y16 (10nM) at the time of reactivation, and the reactivation frequency was compared to untreated controls (reactivation). Reactivation was measured as the frequency of infectious centers determined at 3 weeks post-plating for all groups. Data is shown as the average fold change in infectious centers, as compared to the reactivation group, for four independent donors. Samples were compared by two-way ANOVA followed by Tukey's multiple comparisons test between experimental groups. *P* values are listed for significant comparisons where ** *P* < 0.005 and **** *P* < 0.0001.

### RhoGEFs contribute to viral reactivation *in vivo*

To confirm that US28-mediated activation of RhoGEFs influences viral reactivation *in vivo*, we employed a humanized NSG (huNSG) mouse model previously developed by our group

[47]. In this experiment, huNSG mice were engrafted with human CD34+ HPCs and infected via intraperitoneal injection of TB40/E-GFP-infected fibroblasts as previously described [28]. Because the efficacy of Rhosin was greater than that of Y16 in multiple *in vitro* models, we decided to proceed with only Rhosin for *in vivo* experiments. At 8 weeks post-infection, latently-infected huNSG mice were treated with 40mg/kg Rhosin (N = 5) or DMSO (N = 10). To stimulate viral reactivation, animals in the experimental group were treated with granulocyte-colony-stimulating factor (G-CSF) and AMD3100 (N = 5 Rhosin treated, and N = 5 DMSO treated) while the remaining mice (N = 5 DMSO treated) were left untreated as a control for latency maintenance. One-week post mobilization, spleen tissues were harvested and HCMV viral load was determined by qPCR. Treatment with Rhosin during reactivation resulted in a 73% decrease in viral load, as measured by copies of HCMV viral genomes, when compared with the reactivated but untreated animals (**Fig 7**). Additionally, huNSG mice receiving Rhosin showed comparable viral loads to latently infected animals (**Fig 7**). This data demonstrates that US28-RhoGEF interactions contribute to viral reactivation in an *in vivo* setting in the background of complex-multicellular interaction partners, and highlights the potential of Rhosin to block HCMV reactivation.

## Discussion

The HCMV-encoded chemokine receptor US28 influences several phases of the HCMV lifecycle, including latency and reactivation; however, the exact mechanisms through which US28 functions remains unclear. In the present report, we utilized a proximity-dependent biotinylating enzyme (TurboID) to characterize the US28 interactome under latent and lytic infection modes and focused our study on exploring the role of RhoA and RhoGEFs as an important US28 signaling intermediary. We further explored the relationship between US28 and RhoGEFs *in vitro* where inhibition of RhoGEFs, via the small-molecule compounds Rhosin and Y16, resulted in a substantial decrease in US28-mediated activation of RhoA. Viral latency and reactivation assays utilizing CD34+ HPCs indicated that blocking US28 –RhoGEF signaling pathway results in a significant reduction in infectious virus after exposure to conditions that induce viral reactivation. These findings were recapitulated in a humanized NSG mouse model where treatment with Rhosin prevented the virus from efficiently reactivating. Collectively, our findings indicate that the US28-RhoGEF signaling pathway is required for efficient viral reactivation and provides insight into targets for the development of novel anti-HCMV therapeutics (**Fig 8**).

Classical approaches for the identification of protein-protein interactions have several limitations including altering the cell state or disturbing protein secondary structure [48–50]. Herein, we circumvent these issues by making use of an unbiased biotin ligase system to examine the proteins which directly bind or are in close proximity to US28. Fusing the TurboID enzyme to the C' terminal tail of US28 did not negatively impact signaling in transfected HEK293M cells as measured by SRE and SRF luciferase reporter assays, nor did it appear to alter US28 localization. Our results highlight the utility of BioID-based systems as an efficacious method for characterizing the complete US28 interactome including weak, transient, and indirect interactions. While our findings are consistent with other published US28 proximity labeling studies [39–41], they represent a significant advancement by comparing US28 interactomes between three unique models including transiently transfected cells, lytically infected fibroblasts, and latently infected CD34+ HPCs.

Our group, and others, have previously shown that US28 regulates several cellular processes by binding unique host chemokines (CC vs. CX$_3$C) and activating a number of signaling intermediates (Src, FAK, Pyk2, PLC, IP$_3$, Ras, ERK, PKC, Calmodulin, and RhoA) [15–

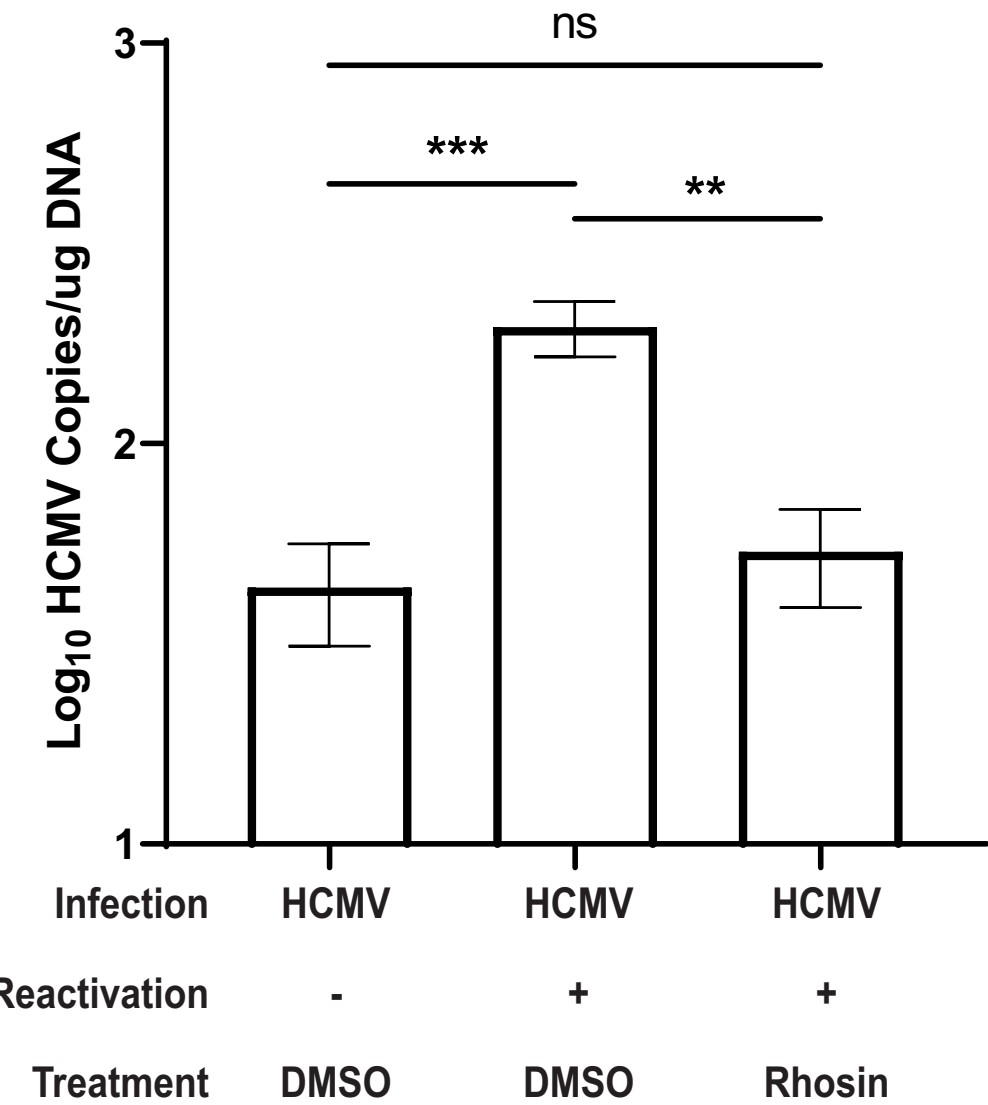

**Fig 7. Rhosin Inhibits HCMV Reactivation *In Vivo*.** Humanized NSG (huNSG) mice were infected with TB40/E-GFP as previously described [28, 47]. After human cell engraftment and viral infection, animals were placed into one of three treatment groups (N = 5 each). At 8 weeks post-infection, one group of huNSG mice (N = 5) was treated with 40mg/kg Rhosin, and the remaining two groups were treated with a comparable volume of DMSO diluent. In parallel, two-thirds of the mice were treated with G-CSF to induce viral reactivation (N = 5 Rhosin treated and N = 5 DMSO treated). Control, latently-infected mice were treated with DMSO but not G-CSF. At 1-week post-treatment (post-reactivation), mice were euthanized and spleen tissues were harvested. Total DNA was extracted using DNAzol, and HCMV viral load was determined by qPCR from two tissue sections per mouse. Error bars represent the standard error of the mean between the average DNA copies per huNSG mouse (N = 5 mice per group). All samples were compared by two-way ANOVA followed by Tukey's multiple comparisons test between experimental groups. *P* values are listed for significant comparisons where ** $P < 0.005$ and *** $P < 0.001$.

22,32,38,41,44]. Consistent with our previous findings, proteomic analyses performed here identified several of these signaling intermediates emphasizing the validity of our BioID system. Interestingly, RhoGEFs were enriched in all three cell culture models. GEFs are responsible for catalyzing the dissociation of GDP from GTPases and provide a direct link between US28 signaling and the downstream RhoA activation cascade. Several studies have implicated RhoA in aspects of the HCMV lifecycle with differential effects dependent on the infected cell

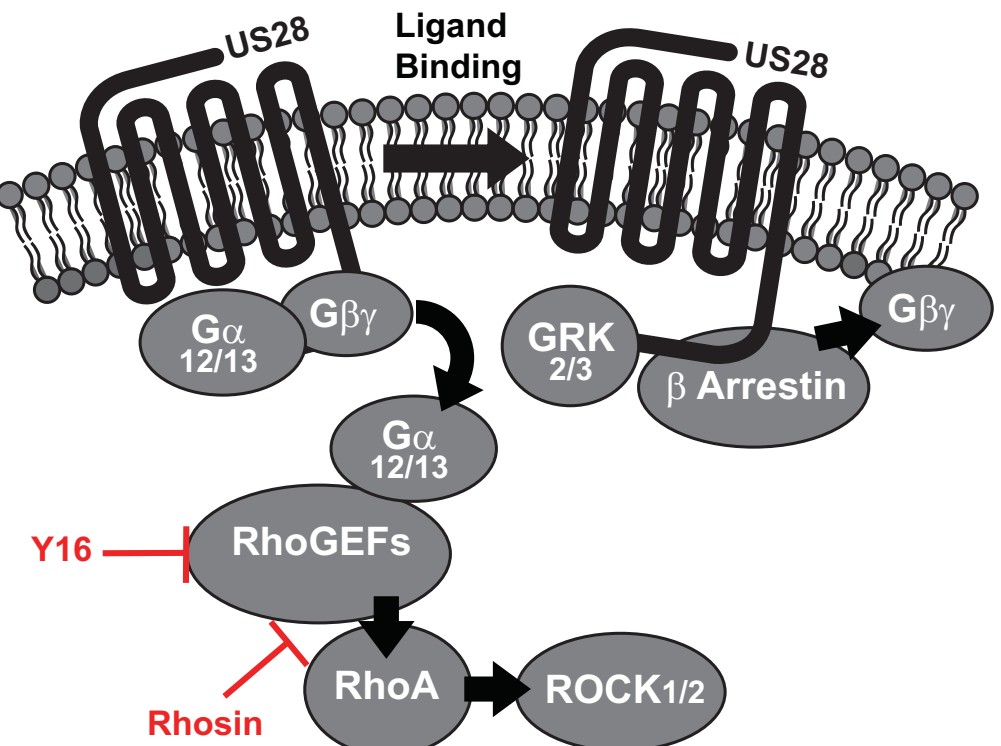

**Fig 8. Model of the US28 –RhoA specific signaling complex depicting RhoGEF interactions and pharmacological small-molecule inhibitors.**

type and chemokine stimulus. For instance, in latently infected CD34[+] HPCs, HCMV miR-US25-1 targets RhoA for downregulation to inhibit cytokinesis and assist in maintaining the viral genome [51]. However, during lytic infection, US28-mediated RhoA signaling facilitates smooth muscle cell migration in response to RANTES [33]. Given these differential effects, we chose to further investigate the role that US28-mediated RhoGEF signaling has on viral reactivation.

We confirmed the physical interaction between US28 and several proteins from our proteomic analysis using a streptavidin bead-based pulldown procedure and traditional immunoblot. To further validate the association between US28 signaling and RhoGEFs, we identified two pharmacological inhibitors of RhoGEFs. Treatment with the small-molecules Rhosin or Y16, both of which target RhoGEF-RhoA interactions, resulted in a significant decrease in US28 signaling in HEK293M cells. Surprisingly, the inhibitory effects of Rhosin treatment were greater than that of Y16 despite targeting similar protein-protein interactions. We hypothesize that this differential effect is largely due to the mechanistic differences between how the two compounds function. Y16 directly blocks the activation of LARG (ARHGEF12), p115-Rho-GEF (ARHGEF1), and PDZ-RhoGEF (ARHGEF11) but not any additional activators of RhoA [45]. Alternatively, Rhosin blocks the entirety of RhoGEF-RhoA interactions by directly binding to two adjacent shallow grooves on the surface of RhoA required for GEF interaction [46]. Therefore, it may be possible that US28 facilitates activation of the RhoA signaling pathway through multiple GEF-RhoA interactions and that greater inhibition of US28 activation of RhoGEFs may be achieved using both compounds synergistically.

Primary CD34[+] HPCs are the gold standard for *in vitro* modeling of HCMV latency and reactivation [52]. Furthermore, huNSG mice represent the only animal model capable of

supporting HCMV infection and we have shown that huNSG mice engrafted with human CD34[+] HPCs to be a reliable and robust model of both latent and lytic infection [47]. Similar to results obtained utilizing a recombinant virus lacking US28 [28], pharmacological inhibition of US28-mediated RhoGEF signaling resulted in a failure of the virus to efficiently reactivate *in vitro* and *in vivo*. We hypothesize that the observed reduction in infectious virus and viral load is due to inhibition of reactivation pathways or an inability of progenitor cells to efficiently traffic out of the bone marrow. This hypothesis is not without precedent as multiple studies have shown RhoGEFs to be required for cellular migration and differentiation in bone marrow-derived cells and macrophages [53–55]. Therefore, these findings provide a direct link to US28-mediated viral reactivation through the RhoA signaling pathway. It is also possible that RhoGEFs interact with other viral proteins such as UL33 and UL78 during viral reactivation. However, these viral GPCRs have not yet been shown to mediate signaling through the RhoA pathway.

The viral GPCR US28 plays an integral role in the pathogenesis of HCMV, establishing differential signaling networks dependent on the presence or absence of bound ligand and the infected cell type. The findings of this study reveal previously unknown US28 interactors which play a crucial role in the facilitation of viral reactivation. To our knowledge, this is the first time specific cellular factors have been implicated in US28-mediated viral reactivation *in vivo*. Additional studies characterizing the US28 interactome in multiple cell types will be required to gain a more comprehensive understanding of the multi-faceted ways in which US28 influences viral latency and reactivation.

## Materials and methods

### Plasmids

Turbo ID was kindly provided by Dr. Alice Ting [56]. US28 and US28-TurboID, containing an in-frame C' terminal fusion with TurboID, were PCR amplified and cloned into pcDNA3.1 (-) (Invitrogen). The PCR fragments were flanked by 5' EcoRI and a 3' HindIII restriction enzyme sites. All clones were transformed into TOP10 Escherichia coli cells (Invitrogen) and confirmed by sequencing. Reporter plasmids pRL-SV40 *Renilla* luciferase (*R*luc), pGL4.33 [luc2P/SRE/Hygro] and pGL4.34[luc2P/SRF-RE/Hygro] containing SRE and SRF responsive elements driving luciferase expression were purchased from Promega.

### Cells and virus

Normal human dermal fibroblasts (ATCC No. PCS-201-010) and human embryonic kidney (HEK) 293M cells (Microbix) were cultured in Dulbecco's modified Eagle's medium (DMEM) supplemented with 10% fetal bovine serum (FBS), penicillin, streptomycin, and glutamine and maintained at 37˚C and 5% $CO_2$. The HCMV strain TB40/E-GFP that constitutively expresses green fluorescent protein under the SV40 promoter [57] was amplified in NHDFs. Infectious virus was determined by limiting dilution plaque assays. The HCMV TB40/E-GFP bacterial artificial chromosome (BAC) was used in a two-step recombination protocol to either replace UL13 with a gH-nLuc reporter cassette (TB40/E-GFP-gHnLuc) or to add an in-frame fusion of TurboID with the C'terminal tail of US28 (TB40/E-GFP-US28-BT) [28]. Following the rescue and expansion of HCMV recombinants, virus preparations were aliquoted and stored at -80˚C. Viral manipulations were confirmed by sequencing. Nano Luciferase activity under the rhesus CMV gH promoter was confirmed and shown to be sensitive to Foscarnet treatment. TurboID expression and activity was confirmed by immunoblotting and biotin labeling.

## Signaling reporter assays

HEK293M cells were plated at $3x10^4$ cells per well in 96-well, white-walled culture dishes. Cells were co-transfected with 50ng pcDNA3.1(-) control, or pcDNA3.1-US28-HA or US28-TurboID along with 10ng of pRL-SV40 (*R*luc) and 50ng pGL4 firefly luciferase reporter vectors (SRE and SRF) using Fugene4K (Promega). At 18 hours post transfection growth medium was replaced with serum-free DMEM with or without small-molecule inhibitors at the indicated concentrations. Luciferase activity was measured in triplicate wells using the Dual Luciferase Reporter Assay System (Promega) at 6 hours post media replacement. Briefly, cell medium was removed and 20μL of passive lysis buffer was added to each well. The 96-well plate was placed at -20˚ C for 30 minutes followed by a 15-minute agitation at room temperature. Luciferase assay reagent was reconstituted and 50μL was injected per well in a Promega GloMax Navigator luminometer for luminescence detection. Assay results were transferred to an Excel spreadsheet, normalized to *Renilla* expression, set relative to the empty vector, and analyzed using GraphPad Prism 9.0 software.

## US28 BioID experiment

For HEK293M experiments, three 10cm cell culture dishes containing 70–80% confluent monolayers of cells were transiently transfected with pcDNA3.1-US28-HA and pcDNA3.1-US28-TurboID-HA. At 18 hours post transfection, HEK293M cells were incubated for 6h in complete media supplemented with 50μg/mL biotin. For NHDF experiments, three 10cm cell culture dishes containing 70–80% confluent monolayers of cells were infected at an MOI of 1 with either HCMV TB40/E-GFP-US28-TurboID, WT HCMV TB40/E-GFP, or mock infected. Three days post infection, cells were incubated for 6h in complete media supplemented with 50μg/mL biotin. In both experiments, cells were scraped, pelleted at 4˚C and washed three times with PBS. Cells were lysed in RIPA buffer (50mM Tris pH 8, 150mM NaCl, 1% triton x-100, 0.1% SDS) and 1x Halt protease inhibitor cocktail (ThermoFisher) and centrifuged at 10,000 relative centrifugal force at 4˚C. Supernatants were incubated with 250μL Pierce NeutrAvidin Agarose beads (ThermoFisher) overnight at 4˚C while rotating. Beads were collected and washed twice for 5 min at 25˚C (all subsequent steps at 25˚C) in 500μL urea wash buffer (PBS pH 7.4, 4M urea). This was repeated three times with wash buffer 2 (PBS pH 7.4, 1% triton x-100), two times with 50mM fresh ammonium bicarbonate and twice with PBS. Bound proteins were removed from the agarose beads with 50μL Laemmli SDS-sample buffer at 42˚C. Twenty-five percent of the sample was reserved for visualizing separated proteins by colloidal Coomassie blue staining and standard immunoblotting. The remaining 75% of the sample (for analysis by mass spectrometry) was washed an additional two times in 50mM ammonium bicarbonate and then resuspended in 268μL 50mM ammonium bicarbonate and incubated on a 70˚C heat block for 10 min with agitation. The samples were immediately treated with 132μL 6M urea and then cooled to room temperature before adding 2.5μL of fresh 0.5M TCEP (Tris(2-carboxyethyl) phosphine hydrochloride; Sigma) and incubated for 30min at room temperature, followed by adding 9μL of fresh 0.5M iodoacetamide and incubating in the dark for another 30 min at room temperature. The samples were then subjected to tryptic digestion by adding 3.7μL 10mM $CaCl_2$ followed by 20μL of 0.1μg/μL sequencing grade trypsin and incubated overnight at 37˚C with rotation. Twenty microliters of formic acid were then added to the eluate and stored at -80˚C until LC-MS/MS analysis. Samples were desalted using ZipTip C18 (Millipore, Billerica, MA) and eluted with 70% acetonitrile/0.1% TFA (Trifluoracetic acid; Sigma) and the desalted material dried in a speed vac. On bead tryptic digests were analyzed by the Fred Hutchinson Proteomics Core Facility (Seattle, WA).

## Orbitrap fusion LC/MS/MS

Desalted samples were brought up in 2% acetonitrile in 0.1% formic acid (12μL) and 10μL of sample analyzed by LC/ESI MS/MS with a ThermoScientific Easy-nLC II nano HPLC system (Thermo Scientific, Waltham, MA) coupled to a tribrid Orbitrap Fusion mass spectrometer (Thermo Scientific, Waltham, MA). Peptide separations were performed on a reversed-phase column (75 μm × 400 mm) packed with Magic C18AQ (5-μm 100Å resin; Michrom Biore-sources, Bruker, Billerica, MA) directly mounted on the electrospray ion source. A 90-minute gradient from 7% to 28% acetonitrile in 0.1% formic acid at a flow rate of 300nL/minute was used for chromatographic separations. The heated capillary temperature was set to 300°C and a static spray voltage of 2100 V was applied to the electrospray tip. The Orbitrap Fusion instru-ment was operated in the data-dependent mode, switching automatically between MS survey scans in the Orbitrap (AGC target value 500,000, resolution 120,000, and maximum injection time 50 milliseconds) with MS/MS spectra acquisition in the linear ion trap using quadrupole isolation. A 2 second cycle time was selected between master full scans in the Fourier-trans-form (FT) and the ions selected for fragmentation in the HCD cell by higher-energy collisional dissociation with a normalized collision energy of 27%. Selected ions were dynamically excluded for 30 seconds and exclusion mass by mass width +/- 10 ppm.

Data analysis was performed using Proteome Discoverer 2.2 (Thermo Scientific, San Jose, CA). The data were searched against Uniprot Human and CRAPome [36] data repositories (>25% cutoff). Trypsin was set as the enzyme with maximum missed cleavages set to 2. The precursor ion tolerance was set to 10 ppm and the fragment ion tolerance was set to 0.6 Da. Var-iable modifications included oxidation on methionine (+15.995 Da), carbamidomethyl on cys-teine (+57.021 Da), and acetylation on protein N-terminus (+42.011 Da). Data were searched using Sequest HT [58]. All search results were run through Percolator for scoring [59].

## Pathway analysis

Reactome pathway analysis software was used to evaluate proteomic data and identify signifi-cantly enriched pathways using the default analysis settings [37]. Significantly impacted canon-ical pathways were also explored using STRING pathway mapping web browser tools to identify and predict additional interactors [60].

## Immunoblot

Cell lysates were harvested in RIPA buffer supplemented with HALT protease inhibitor and stored at -20°C. Proteins were separated on a 4–12% SDS-PAGE gel and blotted on PVDF membranes. Immunoblots were performed using antibodies directed against HA (sc-7392, Santa Cruz Biotechnology), p115 RhoGEF (sc-74565, Santa Cruz Biotechnology), ROCK1 (sc-5562, Santa Cruz Biotechnology), c-Src (sc18, Santa Cruz Biotechnology), β-actin-HRP (sc-47778, Santa Cruz Biotechnology), endothelial cell growth factor receptor (EGFR; Cell Signal-ing; D3881), Streptavidin-HRP (Thermo Scientific; 21130), UL44 (CA006-100, Virusys), and TurboID (AS204440, Agrisera) and if required, with the appropriate HRP conjugated second-ary antibodies (anti-mouse sc-25409 and anti-rabbit sc-2357).

## Microscopy

NHDFs ($1.0x10^5$) were added to each microscope coverslip and maintained at 37°C and 5% $CO_2$. NHDFs were transfected with 1μg of either HA-US28 or HA-US28-BT using Lipofecta-mine 2000 according to manufacturer's instructions. At 16 hours, cells were fixed with 4% paraformaldehyde in PBS, permeabilized in 0.25% Triton-X100 in PBS, and blocked with 2%

bovine serum albumin 0.1% Triton-X100 in PBS. Cells were stained with HA tag monoclonal antibody coupled with Alexa Fluor 488 (1:1,000 dilution; ThermoFisher) and Phalloidin-Alex Fluor 647 (1:1,000 dilution; ThermoFisher) in blocking buffer. Coverslips were washed with PBS and mounted using Fluoromount-G. Images were captured with a Leica Stellaris 8 confocal microscope using Leica Application Suite X software version 4.5.0 (Leica Microsystems).

## Quantitative RT-PCR

RhoGEF gene expression was confirmed by real-time RT-PCR using primer and probe sets for ARHGEF1 (4448892; Hs00180327_m1), ARHGEF11 (4448892; Hs01064532_m1), and ARH-GEF12 (4448892; Hs00209661_m1) available from Thermo Fisher Scientific. Total RNA was isolated using Trizol (Thermo Fisher Scientific) from normal human dermal fibroblasts (NHDF) and CD34+ hemopoietic progenitor cells (HPCs). The RNA was treated with EZ-D-NAse (Thermo Fisher Scientific). cDNA was generated using Superscript IV (Invitrogen) and analyzed by RT-PCR using TaqMan Fast Advanced Master Mix and a QuantStudio 7 Flex Real-Time PCR system. Cycle threshold values were calculated using QuantStudio Design software [61].

## Limiting dilution HCMV latency and reactivation assay

Latency and reactivation was monitored in long-term cultures of CD34+ HPCs using methods as previously detailed [52,62]. Primary CD34+ hematopoietic progenitor cells (HPCs) were isolated using magnetic bead separation (Miltenyi Biotech) and viably frozen as previously described [62]. CD34+ HPCs were thawed and recovered overnight in stem cell media, and infected with HCMV TB40/E-GFP at a multiplicity of infection (MOI) equal to 3 for 48 hours prior to isolation by fluorescence activated cell sorting (FACS) using a FACSAria (BD FACS Aria equipped with 488, 633 and 405nm lasers, running FACS DIVA software) in order to obtain a pure population of viable GFP+ CD34+ HPCs as previously described [52, 62]. The cells were then co-cultured in transwell culture dishes above monolayers of irradiated M2-10B4 and S1/S1 stromal cells. At 14 days post infection (dpi), HPCs were serially diluted in RPMI-1640 medium containing 20% FBS, 2mM L-glutamine, 100U/mL penicillin, 100μg/mL streptomycin, 15ng/mL granulocyte-colony stimulating factor (G-CSF), and 15ng/mL granulocyte-macrophage colony stimulating factor (GM-CSF) and overlaid onto confluent monolayers of NHDFs cultured in 96-well plates. To quantify the levels of pre-reactivation infectious virus, a fraction of the HPC cultures were mechanically disrupted and lysates were serially diluted and then added to NHDFs cultured in 96-well plates. Cell cultures were microscopically visualized for the presence of GFP+ weekly, for up to 4 weeks, to assess the reactivation frequency from latently infected cells and the presence of preformed infectious virus by extreme limiting dilution assay (ELDA) [52].

## Cellular cytotoxicity assay

Compound cytotoxicity was measured following the CellTiter-Glo luminescent cell viability assay (Promega). Briefly, one day prior to the assay, black walled 96-well plates (Corning) were seeded with NHDFs at $1.5 \times 10^4$ cells per well in 50μL. Compounds, starting at a concentration of 40μM, were diluted 1:2 with DMEM supplemented with 5% FBS and 1X PSG. A total of 50μL of diluted compound was added to triplicate wells of the 96-well plate. At 72h following compound addition, 50μL of CellTiter-Glo substrate was added to each well, followed by 2 min on an orbital rocker and a 10 min incubation. The luminescence of each well was measured using a Promega GloMax Navigator luminometer. Well luminescence, indicative of the number of living cells per well, was converted to percent cell viability in Microsoft excel, by

dividing luminescence values in experimental wells by the value in control wells containing untreated cells and multiplying by 100. These values used to calculate compound 50% cellular cytotoxicity ($CC_{50}$) values by nonlinear regression analysis of graphs with compound concentration in log plotted versus cell viability, using GraphPad Prism 9.0 Software.

CD34$^+$ HPCs were differentiated from WA01 human embryonic stem cells using a commercial feeder-free hematopoietic differentiation kit (STEMdiff Heme, Stem Cell Technologies) according to the manufacturer's directions. HPCs were cultured in SFEMII with 10% BIT serum replacement, stem cell cytokines (stem cell factor, FLT3L, IL-3, and IL-6 [PeproTech]), and penicillin/streptomycin, along with increasing concentrations of Rhosin, Y16, or DMSO (control) in triplicate for 5 days. Colorimetric assay (WST-1 based, Roche) was used to perform the cytotoxicity assay according to the manufacturer's directions. Absorbance ($\lambda$420) values were background subtracted from media alone and normalized to DMSO control.

## HCMV nLuc assay

The antiviral activity of Rhosin (Tocris Bioscience) and Y16 (Calbiochem) was measured using the reporter virus HCMV TB40/E-GFP gHnLuc that expresses nanoluciferase under the gH late viral promoter. NHDF cells ($1.5 \times 10^4$ cells/well) were plated in 96-well plates 24 hours prior to start of the assay. Compounds, starting at a concentration of 40μM, were diluted 1:2 with DMEM supplemented with 5% FBS and 1X PSG. Cells were treated with Rhosin, Y16, Foscarnet (positive antiviral control) or DMSO in triplicate and infected with HCMV TB40/E gHnLuc (MOI = 0.3 PFU/cell). At 72 hpi, 50μL Nano-Glo Luciferase Assay reagent (Promega) was added to each well, followed by 2 min on an orbital rocker and a 10 min incubation. Luminescence was measured using a GloMax Navigator Luminometer. Results were graphed using Graphpad Prism 9.0 software.

## RhoA activation assay

NHDFs were plated in 10-cm dishes ($2.0 \times 10^6$ per well) 24 hours prior to treatment with the indicated concentrations of Rhosin, Y16, or an equivalent amount of DMSO (untreated control), diluted in serum-free DMEM for 24 hours. The cells were subsequently stimulated with media containing 10% fetal bovine serum for 15 minutes. Cells were washed once with PBS and lysed in a buffer containing 25 mM Tris-HCl, 150 mM NaCl, 5 mM $MgCl_2$, 1% NP-40, and 5% glycerol. Lysates were clarified by centrifugation at 4˚C and protein concentrations were normalized using the Qubit Protein BR Assay Kit (Invitrogen). Proteins were isolated using the Active Rho Pulldown and Detection kit (Thermo Scientific) according to the manufacturer's recommendations. The total amount of active RhoA was assessed via immunoblot using the indicated primary antibodies. Quantification shows the relative expression of RhoA bound to GTP compared to untreated lysates and normalized to levels of beta actin.

## HCMV infection of humanized mice

Mouse procedures were performed in accordance with approved Institutional Animal Care and Use Committee (IACUC) protocols under the recommendations of the American Association for Accreditation of Laboratory Animal Care (AAALAC). Mice were housed in the Vaccine & Gene Therapy Institute at Oregon Health & Science University vivarium using microisolator cages and fed sterile food and water *ad litem*. For this experiment, humanized mice were generated by irradiating NOD.Cg-*Prkdc*$^{scid}$*IL2R*γ$^{tm1Wjl}$/SzJ (NSG) mice (Jackson Laboratories) by sublethal irradiation of 0- to 3-day-old neonates at 75 cGy using a $^{137}$Cs gamma irradiation source. The irradiated animals were reconstituted by intrahepatic injection of $1 \times 10^5$ human CD34$^+$ HPCs as described previously [28]. Peripheral blood was collected

every 4 weeks beginning at week 8 post-injection to assess human cell engraftment using flow cytometry. 16 weeks post engraftment, mice were distributed to experimental groups normalized for engraftment success as determined by percentage human CD45[+] lymphocytes in the periphery. Humanized mice were dosed with 1 ml of 4% thioglycolate (Brewer's medium; BD) by intraperitoneal (i.p.) injection and then injected i.p. with two T150 flasks of HCMV TB40/E-GFP-infected NHDFs. At 8 weeks post-infection, the animals were divided into three groups. Two groups of latently infected mice were treated with 100 μl of Neupogen (G-CSF; 300 mg/ml; Amgen) by subcutaneous pump and 125μg of AMD3100 administered by i.p. injection to mobilize progenitor cells and promote HCMV reactivation [28,63]. One of the reactivation groups received 40mg/kg Rhosin HCl resuspended in a final volume of 100μL DMSO and the other group was treated with an equivalent amount of DMSO for 7 days by IP injection. The third group of latently infected mice did not receive the reactivation cocktail but was treated with DMSO to serve as comparators for viral levels during latency. At 1-week post mobilization, the mice were euthanized via $CO_2$ administration according to AAALAC euthanasia guidelines, and then blood, bone marrow, spleen, and liver tissues were collected for further analysis.

### Quantitative detection of HCMV viral DNA

Total DNA was extracted from portions of mouse spleen using DNAzol (ThermoFisher) and primers and probe recognizing HCMV UL141 were used to quantify viral genomes by quantitative real-time PCR as previously described [28]. Dilutions of purified HCMV BAC DNA were used to create a standard curve. A 1 μg sample of total DNA was added to each reaction well of TaqMan FastAdvance PCR master mix (Applied Biosystems) and samples were analyzed in triplicate on a StepOnePlus TaqMan PCR machine (Applied Biosystems) with an initial activation at 50˚C for 2 min and 95˚C for 20 s, followed by 40 cycles of 1 s at 95˚C and 20 s at 60˚C. TaqMan results were analyzed using ABI StepOne software and graphed using Prism 9.0 software.

### Supporting information

**S1 Fig. Characterization of RhoGEF Inhibitors Rhosin and Y16. (A)** NHDF cells were treated at the indicated concentrations of Rhosin and Y16 for 24 hours in serum-free media. 24 hours post addition, cells were stimulated with 10% fetal calf serum for 15 minutes. Lysates were harvested and subjected to GST-Rhotekin pulldown according to the manufacturer's recommendations (Thermo Fisher). RhoA activation was assessed via immunoblot using the indicated primary antibodies. Quantification shows the relative expression of RhoA bound to GTP compared to β-actin and normalized to untreated lysates (n = 3, representative blot shown). **(B)** To measure the cytotoxic effects of Foscarnet, Rhosin, and Y16, NHDF cells were treated with dilutions of inhibitors ranging from 40 to 0.078μM or DMSO alone. At 72 hours post-treatment the cells were analyzed using the CellTiter-Glo luminescent cell viability assay according to the manufacturer's (Promega) recommendation. Data is plotted as cell viability relative to the DMSO-treated control cells. Concentrations of the indicated compounds were $Log_{10}$ transformed, and error bars are representative of the standard error of the mean between triplicate experiments. **(C)** CD34[+] HPCs were cultured in SFEMII with 10% BIT serum replacement, stem cell cytokines (stem cell factor [SCF], FLT3L, IL-3, IL-6 [PeproTech]), and penicillin/streptomycin, along with increasing concentrations of Rhosin, Y16, or DMSO (control) in triplicate for 5 days. Colorimetric assay (WST-1 based, Roche) was used to perform the cytotoxicity assay according to the manufacturer's directions. Absorbance ($A420$) values were background subtracted from media alone and normalized to DMSO control. Error bars

represent the standard deviation from two separate experiments. Statistical significance was determined using two-way ANOVA followed by Tukey's multiple comparisons test between experimental groups. **(D)** To investigate the potential antiviral activity of Rhosin and Y16, NHDF cells were pretreated for one hour with 2-fold serial dilutions of inhibitor, ranging from 40 to 0.078µM or DMSO. Treated cells were infected with TB40/E-GFP-gHnLuc at a MOI equal to 0.3. Foscarnet-treated cells were used as a positive control. After 72 hrs incubation, luminescence was quantified using the Nano-Glo Luciferase assay system and measured on a GloMax Navigator microplate reader according to the manufacturer's (Promega) recommendations. Data is plotted as percent inhibition relative to the cells treated with DMSO. Concentrations of the indicated compounds were $Log_{10}$ transformed, and error bars are representative of the standard error of the mean between triplicate experiments.
(EPS)

**S2 Fig. RhoGEF Expression in NHDF and CD34+ HPCs.** Total RNA from uninfected primary CD34+ HPCs (n = 5) and NHDF (n = 3) was obtained using a phenol-chloroform extraction method. Relative expression of **(A)** ARHGEF1 (p115-RhoGEF), **(B)** ARHGEF11 (PDZ-RhoGEF), and **(C)** ARHGEF12 (LARG) was calculated via RT-qPCR relative to β-Actin using the delta-delta CT method.
(EPS)

**S3 Fig. RhoGEF Interactions Contribute to HCMV Reactivation. (A-D)** Representative replicates for experiments performed in Fig 6. Primary CD34+ HPCs isolated from four independent donors infected with TB40/E-GFP (WT-HCMV) were FACS isolated for viable, CD34+, GFP+ HPCs at 2 dpi as previously described [28]. Infected HPCs were cultured on stromal cell support for 12 days to establish latency and equivalent populations of HPCs were co-cultured in supportive media on a fibroblast layer to reactivate virus as previously described [28]. Samples were treated with the RhoGEF inhibitors Rhosin (10nM) or Y16 (10nM) at the time of reactivation and the reactivation frequency was compared to untreated controls (reactivation). Reactivation was measured as the frequency of infectious centers determined at 3 weeks postplating for all groups.
(EPS)

**S1 Table. Pathway Analysis of US28 BirA Interactome in 293 Cells.**
(PDF)

**S2 Table. Pathway Analysis of HCMV-US28 BirA Interactome in NHDFs.**
(PDF)

**S3 Table. Pathway Analysis of HCMV-US28 BirA Interactome in CD34+ HPCs.**
(PDF)

## Author Contributions

**Conceptualization:** Lindsey B. Crawford, Nicole L. Diggins, Meaghan H. Hancock, Andrew Yurochko, Patrizia Caposio, Daniel N. Streblow.

**Data curation:** Samuel Medica, Lindsey B. Crawford, Taylor A. Jones, Timothy Alexander, Sophia Jeng, Shannon McWeeney, Daniel N. Streblow.

**Formal analysis:** Samuel Medica, Lindsey B. Crawford, Chan-Ki Min, Taylor A. Jones, Timothy Alexander, Craig N. Kreklywich, Sophia Jeng, Shannon McWeeney, Meaghan H. Hancock, Patrizia Caposio, Daniel N. Streblow.

**Funding acquisition:** Daniel N. Streblow.

**Investigation:** Samuel Medica, Lindsey B. Crawford, Michael Denton, Chan-Ki Min, Timothy Alexander, Christopher J. Parkins, Gabriel J. Streblow, Adam T. Mayo, Craig N. Kreklywich, Patricia Smith, Meaghan H. Hancock, Patrizia Caposio, Daniel N. Streblow.

**Methodology:** Samuel Medica, Lindsey B. Crawford, Michael Denton, Taylor A. Jones, Timothy Alexander, Christopher J. Parkins, Gabriel J. Streblow, Adam T. Mayo, Craig N. Kreklywich, Patricia Smith, Meaghan H. Hancock, Patrizia Caposio, Daniel N. Streblow.

**Project administration:** Daniel N. Streblow.

**Resources:** Michael S. Cohen, Daniel N. Streblow.

**Supervision:** Shannon McWeeney, Daniel N. Streblow.

**Validation:** Samuel Medica, Lindsey B. Crawford.

**Writing – original draft:** Samuel Medica, Lindsey B. Crawford, Nicole L. Diggins, Meaghan H. Hancock, Daniel N. Streblow.

**Writing – review & editing:** Samuel Medica, Lindsey B. Crawford, Michael Denton, Nicole L. Diggins, Meaghan H. Hancock, Andrew Yurochko, Patrizia Caposio, Daniel N. Streblow.

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
