## [Decision Letter · Decision Letter 0]

7 Feb 2023

Dear Dr. Streblow,

Thank you very much for submitting your manuscript "Proximity-Dependent Mapping of the HCMV US28 Interactome Identifies RhoGEF Signaling as a Requirement for Efficient Viral Reactivation" for consideration at PLOS Pathogens. As with all papers reviewed by the journal, your manuscript was reviewed by members of the editorial board and by several independent reviewers. In light of the reviews (below this email), we would like to invite the resubmission of a significantly-revised version that takes into account the reviewers' comments.

We cannot make any decision about publication until we have seen the revised manuscript and your response to the reviewers' comments. Your revised manuscript is also likely to be sent to reviewers for further evaluation.

Sincerely,

William J Britt

Academic Editor

PLOS Pathogens

Patrick Hearing

Section Editor

PLOS Pathogens

Kasturi Haldar

Editor-in-Chief

PLOS Pathogens

orcid.org/0000-0001-5065-158X

Michael Malim

Editor-in-Chief

PLOS Pathogens

orcid.org/0000-0002-7699-2064

Reviewer's Responses to Questions

**Part I - Summary**

Reviewer #1: The current manuscript from the Streblow group describes a novel mechanism through which viral GCPR US28 – RhoGEF interactions are necessary for viral reactivation from latency. US28 is expressed during latency and regulates a variety of signaling pathways to regulate HCMV latency and reactivation. However, a complete picture of the US28 interactome remains to be elucidated. The authors utilized an unbiased proximity-dependent biotin identification method (TurboID) to determine proteins in close proximity to US28 from which potential interactions with RhoGEFs were identified. The authors take a comprehensive approach to determine if US28 interactions with RhoGEF is required for HCMV reactivation, including in vitro transient transfections and infections with recombinant viruses of cell lines and primary CD34+ stem cells. In vivo reactivation studies were also performed using a huNSG mouse model developed by their group. However, despite the use of parallel approaches, many of the experiments lack important controls that are necessary to support to their conclusions.

Reviewer #2: In the study by Medica and colleagues, the authors have used proximity-dependent biotinylation experiments to identify US28 interacting partners and determined that the US28 signaling pathway converges with the Rho signaling pathway. Experiments were done in both US28 transfected cells and in cells infected with HCMV expressing a US28-BirA fusion protein. siRNA knockdown of Rho-GEFs or pharmacological inhibition of RhoGEF activity blocks US28 dependent signaling as assessed by SRE and SRF reporter assays. The authors go on to assess HCMV reactivation in vitro and in vivo using RhoGEF inhibitors. Inhibition of the Rho pathway in both assays led to a reduction in the efficiency of HCMV reactivation implicating RhoA signaling in the reactivation process. The paper is interesting and well-written, however several major deficiencies exist. First, the reactivation assays are only done with wildtype virus. While the authors do provide some evidence indication that RhoA signaling is required for efficient HCMV reactivation, a direct link between US28 and RhoA dependent signaling leading to viral reactivation is missing. The authors need to include US28 null viruses to draw this conclusion. US28 null viruses should be partially deficient for reactivation and the residual reactivation of US28 null viruses should be unaffected by the Rho inhibitors. Second, there is little mechanistic information presented indicating the mechanism by which US28 engages and/or activates RhoA. Moreover, control renilla luciferase constructs need to be used to solidify the effects of US28 signaling, siRNAs, and pharmacological inhibitors. Third, there are several places where a more granular presentation of data and the corresponding statistical analyses is needed. In particular, the reactivation assays and the effects of the Rho inhibitors is unclear. There are several other areas in the manuscript that could be improved as pointed out below in the specific points. Overall, this manuscript should be further developed for consideration in PLOS Pathogens.

Reviewer #3: In this manuscript, Medica et al. identify proteins that work in conjunction with US28, an HCMV encoded G-protein coupled receptor, to mediate the latent to lytic transition of the virus. There are many strengths to this study, including the extensive BioID data and the use of their humanized mouse model to show that reactivation does not occur in the presence of the RhoGEF inhibitor Rhosin. However, the major claims are not supported by the data and additional work is needed to substantiate these claims.

**Part II – Major Issues: Key Experiments Required for Acceptance**

Reviewer #1: 1. Fig. 1: The authors present data demonstrating that the addition of TurboID does not appear to negatively impact US28 signaling. However, it is important to also demonstrate that the addition of TurboID does not alter US28 localization. This is a critical control given that localization could significantly impact the US28 interactome.

2. Given that HEK293M cells are not permissive for HCMV replication, the authors should provide a justification as to why these cells were chosen for this study.

3. Fig. 3a: US28 interactions were validated from HEK293M cell transfected by a HA-tagged US28-TurboID protein (US28-BT-HA) construct. However, these interactions should be validated in the context of a viral infection in NHDF (or another permissive cell line) as the authors have already generated a HCMV TB40E-GFP-TurboID (BT).

4. Fig. 3B lacks statistics. The labelling is also unclear. Presumably ARCHGEFs represent siRNAs against different RhoGEFs? TIRO needs to be defined in the figure legend. There is no scrambled control siRNA treatment group. The authors provide no evidence that the siRNA worked or how efficient the knockdown was.

5. Fig 3C: They authors use 2 Rho inhibitors but provide no evidence that either inhibitor is working as expected in their model system.

6. Supplementary data appears to be mislabeled and/or missing. Cytotoxicity studies (Fig. S2A) could not be found.

7. Fig. 4. Although the authors validate US28 interactors with transient transfections in HEK293M cells, US28 interaction with RhoGEFs needs be validated in the context of latency. This may be difficult with CD34+ cells, however THP-1 cells, which allow for HCMV latency and the expression of US28 during latent infection, could be used.

8. Fig. 5: The authors provide evidence that treatment with Rhosin prevents HCMV reactivation in vivo based on HCMV copy number following G-CSF treatment. Is it not possible that Rhosin simply attenuates HCMV replication, which leads to lower HCMV copy numbers? The authors do state in Fig.S2B (which could not be found), Rhosin had no effect on HCMV replication in vitro. However, this may not be the case in vivo. Mice could be treated with Rhosin following G-CSF reactivation to demonstrate that Rhosin has no effect of viral replication rates in vivo.

9. Overall, the authors conclude that blocking US28-RhoGEF interactions prevents HCMV reactivation. Although it is clear that RhoGEF signaling is required, the authors do not provide direct evidence that blocking US28-RhoGEF interactions prevents viral reactivation. At least showing that Rhosin (a steric inhibitor) blocks US28 interaction with RhoGEFs would provide support that the interaction is necessary for HCMV reactivation.

Reviewer #2: 1. For the reporter assays in Figure 1C and 1D, were internal renilla luciferase control levels measured and used to correct the SRE and SRF reporter data? Could the data be recalculated and shown as a fold over the basal (empty vector)? As shown it is difficult to ascertain the level of US28 based signaling compared to the empty vector?

2. A more granular and easy to follow analysis of the comparison between transfected and infected cell US28 interactome would be useful. While Figure 2 attempts to do this, it is not sufficient. As an example in the TB40E interactome in 2C, Rho GTPase Cycle is listed twice with different number of proteins, while 2A and 2C differ with 2A listing RhoA GTPase effectors. Both list “Signaling by Rho GTPases” but it is not clear if these represent the same or significantly overlapping proteins.

3. While the pulldowns in Figure 3A look clean and interacting proteins are absent in the non-BirA expressing cells, an additional control showing these pull downs in cells expressing an irrelevant BirA expression vector would be more convincing. The authors only look at p115-RhoGEF, have the authors done similar experiments with other Rho-GEFs.

4. Mechanistic information regarding how interaction between US28 and the Rho-GEFs leads to activation of Rho is lacking. Does US28 cause a relocalization of Rho-GEF that facilitates GTP loading onto Rho? Does US28 cause an increase in GTP bound Rho? Can the requirement for US28 in reactivation become overcome by indirectly activating Rho or by expressing constitutively active mutants of Rho?

5. In Figure 3B, it is not clear which RhoGEF is targeted by the indicated siRNAs. Are they all p115-RhoGEF specific? Moreover, there are no error bars and it is not clear as to how many times this experiment was repeated. In Figure 3C, the effects of Y16 and Rhosin seem significant. However, it is again unclear if an internal constitutive renilla reported was used to control for the firefly luciferase and if the pharmacological inhibitors could have in fact blocked both the reporter and the constitutive renilla construct.

6. In Figure 4A the results of one experiment are shown, while in Figure 4B the combined results of 4 independent donors are shown. In Figure 4B the results are presented as fold change relative to the reactivation control, which in each experiment is set to 1.0. This eliminates the variability of the reactivation control and I am not certain that the statistics are vaild. I can appreciate that the frequency of infectious centers would be variable from donor to donor, which is why the data is set to 1.0 for the reactivation control. Perhaps the data in Figure 4B can be superimposed with the individual data points from the 4 experiments or the actual average frequency of infectious centers could be displayed for the 4 independent experiments.

7. The experiments in Figure 4 should be shown with both wildtype and US28 null viruses. Since the authors are claiming that US28 signaling through Rho is important for reactivation it would be important to show the levels of reactivation for US28 null virus as well as the effects of the inhibitors on reactivation of US28 null virus. Moreover, in vivo experiments in Figure 5 assessing the effects of Rhosin on inhibiting reactivation should similarly be performed with both wildtype and US28 null viruses.

Reviewer #3: 1. A major concern is the overinterpretation of the BioID data. As properly stated in lines 50, 98, 163 and 488, this assay detects proximal proteins. As shown in Figure 1A, this could include interacting proteins or those simply nearby. The terms interaction(s), interactome and interactor are used heavily throughout the manuscript (40, 41, 55, 96, 110, 112,114, 127, 148, 159, 161, 170, table 1, etc.), and even in the title. However, no work has been done to confirm whether any of these hits are actually interacting with US28, as opposed to just in proximity to US28. In fact, the statement on line 267 “We confirmed the physical interaction between US28 and several proteins from our proteomic analysis” is actually false. Given that the proteins were pulled down with NeutrAvidin beads, this experiment confirmed only that they were biotinylated, meaning in close proximity to US28, not that they were physically interacting. These hits should be investigated for a physical interaction using co-immunoprecipitations (or other appropriate assay).

2. Given the differences observed between transfection and infection, particularly with the STRING analysis, the relevancy of the transfection data is questionable with respect to making any conclusions about infection. The results in figure 3B would therefore be better in the context of infection, similar to how the results of 3C were shown for infection in Figure 4. Even if signaling is decreased following knockdown of the targets after transfection, this may or may not be applicable in the context of the virus. An additional concern with Figure 3B is the lack of any confirmation that the respective targets were knocked down. This is an essential piece of data for interpreting the results and a western blot should be included.

3. The use of the humanized mouse model to show that inhibition of Rho-GEFs via Rhosin prevents reactivation of latency is a strength of this paper. However, the conclusions of this experiment are overstated. Line 227-228 states that “This data demonstrates that US28-RhoGEF interactions contribute to viral reactivation.” The results do show a role for RhoGEFs in reactivation, which is inhibited by Rhosin, but the link to US28 is assumed. Other pathways involving RhoGEFs could equally be involved. No conclusions can be made about US28 from the data as currently presented. This conclusion requires breaking the US28-RhoGEF link, not just inhibiting the Rho-GEFs. One way to show this could involve identifying mutations on US28 that break that link.

**Part III – Minor Issues: Editorial and Data Presentation Modifications**

Reviewer #1: There are some minor typos throughout the manuscript.

Reviewer #2: 1. Supplemental Figure 2A appears to show the data as a log scale, but it the text the authors refer to drug concentrations up to 40uM. This could be shown more clearly. Moreover, although the drugs are not toxic to fibroblasts, the authors have not assessed toxicity in HPCs, which would be more relevant to the current study, which analyzes the effects of these drugs on HCMV reactivation in HPCs.

2. The RT-qPCR data should also indicate the more common Rho-GEF nomenclature (p115-RhoGEF, PDZ-RhoGEF, etc.) in addition to the actual gene name.

Reviewer #3: 1. The source of the cells, either 293M or NHDF, are not included in the Materials and Methods

2. For Supplemental Figure 3, there is a clear difference in transcript levels between cells as can be deduced from Ct values. However, a standard curve would allow for quantification and provide actual numbers for the differences in transcript levels between the two cell types.

3. Figure 4A shows the data for one HPC donor. It would be nice to include in Figure 4A the data from all four donors that was used to calculate data for Figure 4B to allow for a sense of variability amongst donors.

4. Figure 6 contains inactive and active versions of US28, but there is no discussion of the different US28 versions at all in the text. Searching for the term inactive returns no hits. Please provide an explanation in the text or adjust the figure appropriately.

5. Line 234 states that “Our findings are consistent with a recently published US28 proximity labeling study”, yet many of the top hits between the two studies do not appear to overlap. Was EphA2 found in your study? That study identified only ARHGEF17, but not other RhoGEFs. There should be more discussion on the similarities and the differences between the two studies, and the claim on line 234 is a bit misleading.

PLOS authors have the option to publish the peer review history of their article (what does this mean?). If published, this will include your full peer review and any attached files.

Reviewer #1: No

Reviewer #2: No

Reviewer #3: No
---

## [Decision Letter · Decision Letter 1]

7 Aug 2023

Dear Dr. Streblow:

Thank you very much for submitting your manuscript "Proximity-Dependent Mapping of the HCMV US28 Interactome Identifies RhoGEF Signaling as a Requirement for Efficient Viral Reactivation" for consideration at PLOS Pathogens. As with all papers reviewed by the journal, your manuscript was reviewed by members of the editorial board and by several independent reviewers. The reviewers appreciated the attention to an important topic. Based on the reviews, we are likely to accept this manuscript for publication, providing that you modify the manuscript according to the review recommendations.

There is a consensus by the reviewers that this manuscript is acceptable for publication in PLOS Pathogens; however, both reviewers and in particular reviewer 2, have provided several constructive modifications that will improve the overall presentation of the data from this study. Incorporation of these modications will benefit this manuscript and should be accomplished prior to formal acceptance. Both revirewers agree that the manuscript provides significant new information of relevnce to the biology of CMV.

Sincerely,

William J Britt

Academic Editor

PLOS Pathogens

Patrick Hearing

Section Editor

PLOS Pathogens

Kasturi Haldar

Editor-in-Chief

PLOS Pathogens

orcid.org/0000-0001-5065-158X

Michael Malim

Editor-in-Chief

PLOS Pathogens

orcid.org/0000-0002-7699-2064

There is a consensus by the reviewers that this manuscript is acceptable for publication in PLOS Pathogens; however, both reviewers and in particular reviewer 2, have provided several constructive modifications that will improve the overall presentation of the data from this study. Incorporation of these modications will benefit this manuscript and should be accomplished prior to formal acceptance. Both revirewers agree that the manuscript provides significant new information of relevnce to the biology of CMV.

Reviewer Comments (if any, and for reference):

Reviewer's Responses to Questions

**Part I - Summary**

Reviewer #1: The manuscript is significantly improved from the original version. The authors have provided new data that substantially improve the rigor of the study and provide the evidence necessary to support their conclusions. Overall, the investigators have adequately addressed all of my concerns.

Reviewer #2: The HCMV US28 GPCR has emerged as an important viral gene involved in a number of processes including latency/reactivation, smooth muscle cell migration, etc. US28 has been shown to function as an authentic GPCR, coupling to multiple G-proteins. However, the full repertoire of cellular proteins that US28 interacts with is largely unknown and more information in this regard is clearly needed. The current study uses a proximity labelling approach to identify US28 interacting proteins in both an overexpression system as well as in HCMV infected fibroblasts and hematopoietic progenitor cells. A number of interesting pathways are identified and the authors focus on the Rho pathway, showing that it plays a role in latency and reactivation. The authors have responded favorably to my earlier concerns and I am now in support in principal of acceptance of this manuscript. I have a couple of minor points that need to be addressed prior to publication.

**Part II – Major Issues: Key Experiments Required for Acceptance**

Reviewer #1: None

Reviewer #2: None

**Part III – Minor Issues: Editorial and Data Presentation Modifications**

Reviewer #1: 1) Although the western blots are clear and support their conclusions, the figure legends should include the number to biological replicates that were performed for each immunoblot.

2) Error bars appear to missing in Figure 5B.

Reviewer #2: 1. In Figures 2C and 2D, the Streptavidin-HRP blots are from whole cell extracts (unbound), not pull downs, correct? It is curious that there are far more proteins labelled in the infected NHDFs and CD34+ cells than in the HEK cells. The authors should comment on this or clarify.

2. It is interesting that US28 shows up in the list of viral interacting proteins in NHDFs, but not CD34+ cells. One would think that the BirA would biotinylate the US28 that it is fused with and then show up in both NHDFs and CD34+ cells. The authors should comment on this or clarify.

3. In Figure 5B, the SRF reporter assay data does not show statistical significance or error bars. In the text of the manuscript these data are interpreted as a “marked decrease of US28 signaling”. Moreover, from the figure legend it is not apparent if these data were corrected with renilla luciferase as an internal control?

4. Supplemental Figure 1 labelling is inconsistent between the text, legend, and figure. This needs to be corrected.

5. The authors should carefully review all figure panels to ensure that all legends and panels match the text.

6. Since Figure 6 is an average of the 4 independent donors (which are shown in Figure S3), this should be mentioned in the text in the results section. If possible, the individual data points from Figure S3 could be overlayed onto the data in Figure 6, which would help to better represent the variability between individual donors/experiments.

PLOS authors have the option to publish the peer review history of their article (what does this mean?). If published, this will include your full peer review and any attached files.

Reviewer #1: No

Reviewer #2: No

Figure Files:

Data Requirements:

Reproducibility:

References:

---

## [Editor Report · Decision Letter 2]

12 Sep 2023

Dear Dr. Streblow,

We are pleased to inform you that your manuscript 'Proximity-Dependent Mapping of the HCMV US28 Interactome Identifies RhoGEF Signaling as a Requirement for Efficient Viral Reactivation' has been provisionally accepted for publication in PLOS Pathogens.

Best regards,

William J Britt

Academic Editor

PLOS Pathogens

Patrick Hearing

Section Editor

PLOS Pathogens

Kasturi Haldar

Editor-in-Chief

PLOS Pathogens

orcid.org/0000-0001-5065-158X

Michael Malim

Editor-in-Chief

PLOS Pathogens

orcid.org/0000-0002-7699-2064
---

## [Editor Report · Acceptance letter]

18 Sep 2023

Dear Dr. Streblow,

We are delighted to inform you that your manuscript, "Proximity-Dependent Mapping of the HCMV US28 Interactome Identifies RhoGEF Signaling as a Requirement for Efficient Viral Reactivation," has been formally accepted for publication in PLOS Pathogens.

Best regards,

Kasturi Haldar

Editor-in-Chief

PLOS Pathogens

orcid.org/0000-0001-5065-158X

Michael Malim

Editor-in-Chief

PLOS Pathogens

orcid.org/0000-0002-7699-2064